PHR | SSPH+ SWISS SCHOOL OF PUBLIC HEALTH

published: 02 June 2021
# Community-Based Interventions for Cardiovascular Disease Prevention in Low-and Middle-Income Countries: A Systematic Review

Rawlance Ndejjo [1,2]*, Hamid Yimam Hassen [2], Rhoda K. Wanyenze [1], David Musoke [1], Fred Nuwaha [1], Steven Abrams [2,3], Hilde Bastiaens [2†] and Geofrey Musinguzi [1,2†]

[1]Department of Disease Control and Environmental Health, School of Public Health, College of Health Sciences, Makerere University, Kampala, Uganda, [2]Department Family Medicine and Population Health, Faculty of Medicine and Health Sciences, University of Antwerp, Antwerp, Belgium, [3]Data Science Institute, Interuniversity Institute for Biostatistics and Statistical Bioinformatics (I-BioStat), UHasselt, Belgium

**Edited by:**
*Ana Ribeiro,*
*University Porto, Portugal*

**Reviewed by:**
*Marta Fadda,*
*University of Italian Switzerland,*
*Switzerland*
*Pablo Perel,*
*University of London, United Kingdom*
*Louise Hartley,*
*RTI Health Solutions, United Kingdom*

**\*Correspondence:**
*Rawlance Ndejjo*
*rndejjo@musph.ac.ug*

†*These authors have contributed equally to this work*

**Citation:**
*Ndejjo R, Hassen HY, Wanyenze RK, Musoke D, Nuwaha F, Abrams S, Bastiaens H and Musinguzi G (2021) Community-Based Interventions for Cardiovascular Disease Prevention in Low-and Middle-Income Countries: A Systematic Review. Public Health Rev 42:1604018. doi: 10.3389/phrs.2021.1604018*

**Objectives:** To synthesize evidence on the effectiveness of community-based interventions for cardiovascular disease (CVD) prevention in low- and middle-income countries (LMICs) to inform design of effective strategies for CVD prevention.

**Methods:** We searched MEDLINE, EMBASE, CINAHL, Cochrane register of controlled studies and PSYCINFO databases for studies published between January 2000 and June 2019. Other studies were identified from gray literature sources and review of reference lists of included studies. The primary outcomes for the review were those aimed at primary prevention of CVD targeting physical activity, diet, smoking and alcohol consumption.

**Results:** Database searches yielded 15,885 articles and 94 articles were identified through snowball searching. After screening, the articles from LMICs were 32 emanating from 27 studies: 9 cluster randomized trials, eight randomized controlled trials and 10 controlled before and after studies. Community-based interventions successfully improved population knowledge on CVD and risk factors and influenced physical activity and dietary practices. Evidence of interventions on smoking cessation and reduced alcohol consumption was inconsistent.

**Conclusion:** This evidence should inform policy makers in decision-making and prioritizing evidence-based interventions.

**Keywords:** smoking, cardiovascular disease, knowledge, physical activity, alcohol, community-based, diet, effectiveness

---

**Abbreviations:** CVD, Cardiovascular disease; LMICs, Low- and middle-income countries; NCDs, Non-communicable diseases; PRISMA, Preferred Reporting Items for Systematic Review and Meta-Analyses; RoB, Risk of Bias; ROBINS-I, Risk of Bias in Non-randomized Studies of Interventions; SPICES, Scaling up Packages of Interventions for Cardiovascular disease prevention in selected sites in Europe and sub-Saharan Africa.

# INTRODUCTION

Cardiovascular disease (CVD) continues to disproportionately cause morbidity and mortality in low- and middle-income countries (LMICs). Of the 17.9 million CVD related deaths reported worldwide in 2016, 75% occurred in LMICs [1]. In many LMICs, epidemiological transition, industrialization, infectious diseases burden and globalization have influenced changes in lifestyle observed through changes in physical activity, diet, alcohol and smoking behavior among others [2]. These lifestyle changes have contributed to the upsurge in CVD metabolic risk factors such as obesity, hypertension and diabetes [2]. It is estimated that over half of the 671 million obese population in the world live in 10 countries, eight of which are LMICs [3]. Moreover, the number of people living with diabetes in LMICs is estimated to rise to 228 million by 2030 from 84 million in 1995 [4]. A 2015 systematic review reported a pooled prevalence of hypertension in LMICs of 32.3% (95% CI: 29.4–35.3) [5] while a prevalence of 57.0% (95% CI 52–61%) was reported in another review among the African population aged 50 years and above [6].

Targeting lifestyle factors such as physical inactivity, poor diets, smoking and alcohol intake, and metabolic risk factors including dyslipidemia, hypertension and diabetes can reduce the overall burden of CVD [7]. Community-based interventions target change among individuals, groups, and organizations to avoid development of CVD risk factors or control them and often incorporate strategies to create policy and environmental changes [8, 9]. Through community-based interventions, reduction in CVD burden and risk can be achieved in entire communities impacting population level knowledge and perceptions and risk reduction practices [10, 11]. Population level public health measures are also likely to be more cost effective [12] than treatment oriented programmes for which most LMICs lack capacity to implement on a large scale [13].

Although community-based interventions aimed at CVD prevention have been implemented in LMICs, gaps remain regarding their effectiveness in these settings. Previous reviews on community based interventions have not been specifically directed to LMICs [14, 15], evaluated only a few of the interventions or outcomes [16–18], or do not include recent evidence [19]. This systematic review was aimed at providing up-to-date and comprehensive evidence on the effectiveness of community-based interventions for CVD prevention to support the design of effective strategies for CVD prevention. This review therefore answers two key research questions considering LMICs:

1) What community-based interventions and strategies have been implemented for CVD prevention in LMICs?
2) What is the effectiveness of community-based interventions for CVD prevention in LMICs?

# METHODS

This study was conducted and reported in accordance with the Preferred Reporting Items for Systematic reviews and Meta-Analysis Protocols (PRISMA-P) 2015 statement (S1 Checklist) [20]. The study protocol was registered in the Prospero International prospective register of systematic reviews (Registration Number: CRD42019119885).

## Eligibility Criteria

This review included studies conducted between January 2000 and June 2019 to obtain recent relevant evidence on community-based interventions for cardiovascular disease prevention to inform policy and practice applicable in the current social dynamics. Studies were included if they met the criteria below:

- Study population: Studies conducted among adults aged 18 years and above.
- Intervention: Studies that reported interventions carried out within the community for either primordial or primary prevention of CVD aimed to improve cardiovascular risk knowledge and healthy lifestyle such as physical activity, healthy dietary habit, cessation of smoking and alcohol consumption. These interventions include health education and promotion, community mobilization, lifestyle counseling and coaching, screening, and treatment. The studies ought to have been implemented in a community setting including households, workplaces, schools, sport centers, pharmacies, primary health care units, community health worker posts among others but not secondary care health facilities. Interventions that started at health facilities and later linked to the community were included.
- Comparator: studies where intervention was compared with another intervention, usual care or nothing.
- Outcomes: The primary outcomes were changes in knowledge regarding CVD, physical activity, diet, smoking, and alcohol consumption. Among studies that had at least one primary outcome, secondary outcomes of body weight, systolic and diastolic blood pressure, blood glucose and lipid levels were reported. Studies among patients with CVD conditions and those whose aim was not to prevent CVD or their risk factors were excluded.
- Study designs: Individual level or cluster randomized controlled trials, controlled before and after and controlled interrupted time series studies.
- Context: This was a broad review that was not restricted to any geographical location. However, this article includes only filtered studies conducted in LMICs as defined by the World Bank Gross National Income per capita, calculated using the World Bank Atlas method as of June 2019.
- Language: This review was restricted to articles published in the English language.
- Other considerations: We included only studies that had a sample size of at least 150 participants, a follow-up period of at least nine months and a participant attrition rate of less than 40% to minimize bias from included studies. We also excluded duplicate publications, systematic or narrative reviews, reviews, abstracts, letters to the editor, comments, case reports, conference presentations and study protocols.

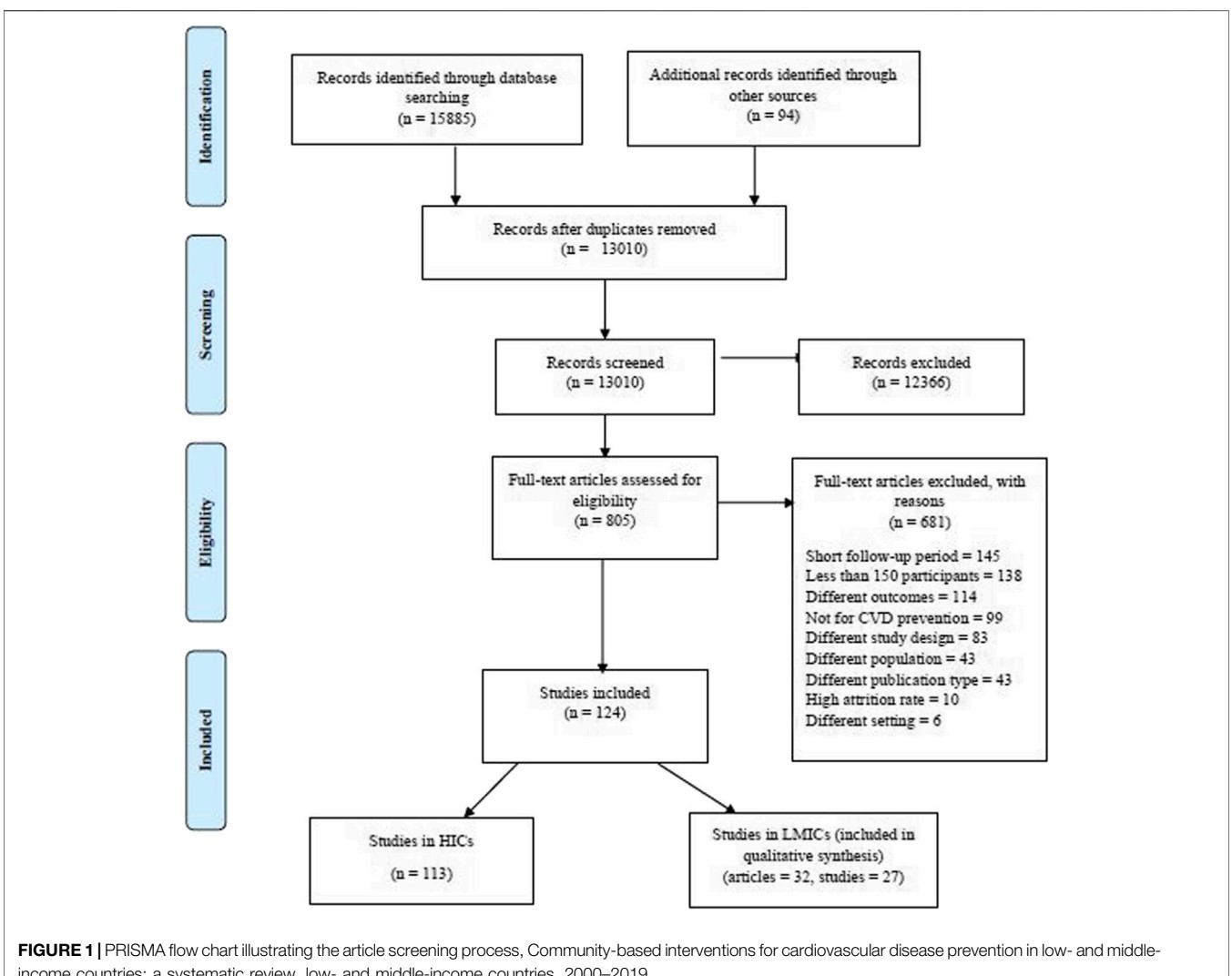

**FIGURE 1 |** PRISMA flow chart illustrating the article screening process, Community-based interventions for cardiovascular disease prevention in low- and middle-income countries: a systematic review, low- and middle-income countries, 2000–2019.

## Information Sources and Search Strategy

We searched MEDLINE, EMBASE, CINAHL, Cochrane register of controlled studies and PSYCINFO. Other sources of publications including thesis online, OpenGrey, ProQuest, Google Scholar and the World Health Organization (WHO) International Clinical Trials registry platform were searched. In addition, we searched reference lists of included studies and similar systematic reviews for potential eligible studies to include in this review. A comprehensive search strategy relating to the population, intervention, and outcomes was developed in MEDLINE (S2 Search strategy) and adjusted to suit other databases. Appropriate limits were applied and only studies conducted between January 2000 to June 2019 were retrieved. The search was however repeated before submission of the systematic review article to include any newer articles. Articles from all comprehensive searches of databases and gray literature and those obtained from reference lists of other articles were exported as EndNote files (including titles and abstracts) and then imported into EndNote as a single library. Duplicate articles from the searches were verified and removed. The remaining articles

were imported into rayyan.QCRI.org [21], a web-based tool that facilitates screening and collaboration among researchers, for screening.

## Screening and Data Extraction

Screening was conducted at two levels, title and abstract and full text independently in rayyan. QCRI by two reviewers (RN and HH) using defined criteria and in case of any disagreements, a third reviewer (HB or GM) made the final decision. We contacted authors by email in case of any key missing information in the articles. All reasons for exclusion of articles were noted and the review process was summarized within the PRISMA flow chart (**Figure 1**) highlighting the process of screening articles [22]. Data extraction was also independently conducted by two of the authors (RN and HH), thereby extracting all relevant information from included full text articles into a standardized Excel spreadsheet and later comparing and resolving any discrepancies. Data were extracted on study and participant characteristics, context, study design, methodology, intervention characteristics, comparator group(s) and outcome

measures. For the outcomes, any effect estimates and observed changes in knowledge about CVDs, uptake of physical activity and diet, or reduction in smoking and alcohol use were recorded and data on secondary outcomes also extracted. Results of studies presented in multiple papers for the same population were only included in the review once.

## Risk of Bias Assessment

The risk of bias of included studies was assessed using the revised Cochrane tool for Risk of Bias (RoB2) for randomized studies [23] and the Risk of Bias In Non-randomized Studies–of Interventions (ROBINS-I) for non-randomized studies [24]. The risk of bias assessment was conducted independently by two reviewers (RN and HH) who resolved any differences through consensus and if necessary, after consultation with a third reviewer (HB or GM).

## Strategy for Data Synthesis

Data for this review were synthesized narratively while answering the aforementioned research questions. Findings have been descriptively presented and discussed while elaborating about the interventions and primary and secondary outcomes. Data have been presented in tabular form for comparison highlighting country, year of study, study objective, intervention, context, population and outcomes among others.

## RESULTS

## Search Results

Databases searched for this review yielded 15,885 articles. On top of that, 94 articles were identified through gray literature and snowball searching. At title and abstract screening, we retained 805 articles that underwent full text screening yielding 124 articles that met the inclusion criteria. Of the 124 articles, 32 were from LMICs representing 27 studies and were included in this review. A flow chart including details of the article screening process is shown in **Figure 1**.

## Characteristics of Included Studies
### Study Setting

Of the 32 articles from which data were extracted, five were from the Isfahan Healthy Heart Program [25–29] and two from The Tehran Lipid and Glucose Study [30, 31] both conducted in Iran, which are reported using the most recent reference as [26, 30] respectively. Thus, this review includes a total of 27 studies as five articles were secondary references. Fourteen studies were from lower-middle [32–45] and eleven from upper-middle income countries [26, 30, 46–54] or both [55] and only one was from a low-income country [56]. One study was conducted in Kenya in sub-Saharan Africa [32] and another in three countries, China, India and Mexico [55]. Of the remaining studies, seven were from India [33–37, 43, 55], five from China [46–50], and two from Iran [26, 30], Pakistan [38, 39] and Sri Lanka [42, 44]. Bangladesh [40], Grenada [54], Malaysia [51], Nepal [56], Thailand [53], Russia [52] and Vietnam [41] each had a single study.

## Study Designs and Context

Nine studies were cluster randomized controlled trials [33, 34, 39, 40, 42–45, 56], eight were randomized controlled trials [35–37, 46, 48, 50, 53, 54] and ten were controlled before and after studies [26, 30, 32, 38, 41, 47, 49, 51, 52, 55]. Studies were carried out in different contexts; eight in urban [30, 32, 36, 37, 39, 48, 50, 56], three in semi-urban [42, 44, 51], eight in rural [33, 35, 40, 41, 43, 45, 49, 53] and two in both rural and urban areas [26, 38]. In six studies, the context was unclear [34, 46, 47, 52, 54, 55].

## Risk of Bias

Among the included studies, the risk of bias categorization was: low (7), moderate (7), some concerns (9) and high/serious (4). The major sources of bias within included studies were: outcome measurement bias due to limited use of objective measures or validated tools, missing data bias due to omission of missing data or not applying appropriate analytical techniques, and the lack of control for some confounders for non randomized studies. **Figures 2**–**4** show the risk of bias graphs and their summary drawn using the visualization tools by McGuinness and Higgins [57]. Overall, the randomized controlled and cluster randomized controlled trials had a lower risk of bias compared to the controlled before and after studies.

## Study Population

Most studies targeted whole populations [26, 30, 32, 33, 38, 40–45, 47, 49, 55, 56] with health promotion and disease prevention activities while a few targeted risk groups including the elderly [48, 50], smokers [35, 52], individuals at high risk of diabetes [34, 36, 37, 51], those with hypertension [39, 46, 53] or more than one CVD risk factor [54]. All studies targeted both males and females except for four, three conducted in India, that targeted only males [35–37], and one in Sri Lanka that targeted mothers [54]. The age of participants in the studies was from 18 years and above and participant numbers ranged from 297 in Malaysia [51] to 12,514 in Iran [26].

## Variety of Community-Based Interventions

The community-based interventions involved health education and awareness creation through mass media, mobile phones as well as information, communication and education materials [26, 30, 32, 33, 35–41, 45–47, 49, 50, 55]; trainings through workshops, lectures and small groups [26, 30, 38, 39, 48–51, 56]; lifestyle consultation and counseling either individually or in groups [33, 35–37, 39, 48, 51, 53, 56]; and community mobilization activities through meetings, peer support programmes and competitions [26, 30, 32, 34, 40, 41, 45, 47, 51, 52, 55]. The other interventions were: environmental and structural changes in policies, infrastructure or institution of restrictions [26, 30, 47, 55]; and screening and treatment of risk factors [26, 32, 36, 41, 42, 48, 53, 56]. All studies used more than one strategy and most involved sharing information, health education, provision of community services and social mobilization. The least used strategies were changes in organisational culture and health policy and enforcement.

The intervention were delivered by healthcare workers [26, 30, 32, 36, 39, 41, 47–50, 52, 53], community health workers, peers and volunteers [26, 30, 32–34, 38–40, 42, 53], local leaders and

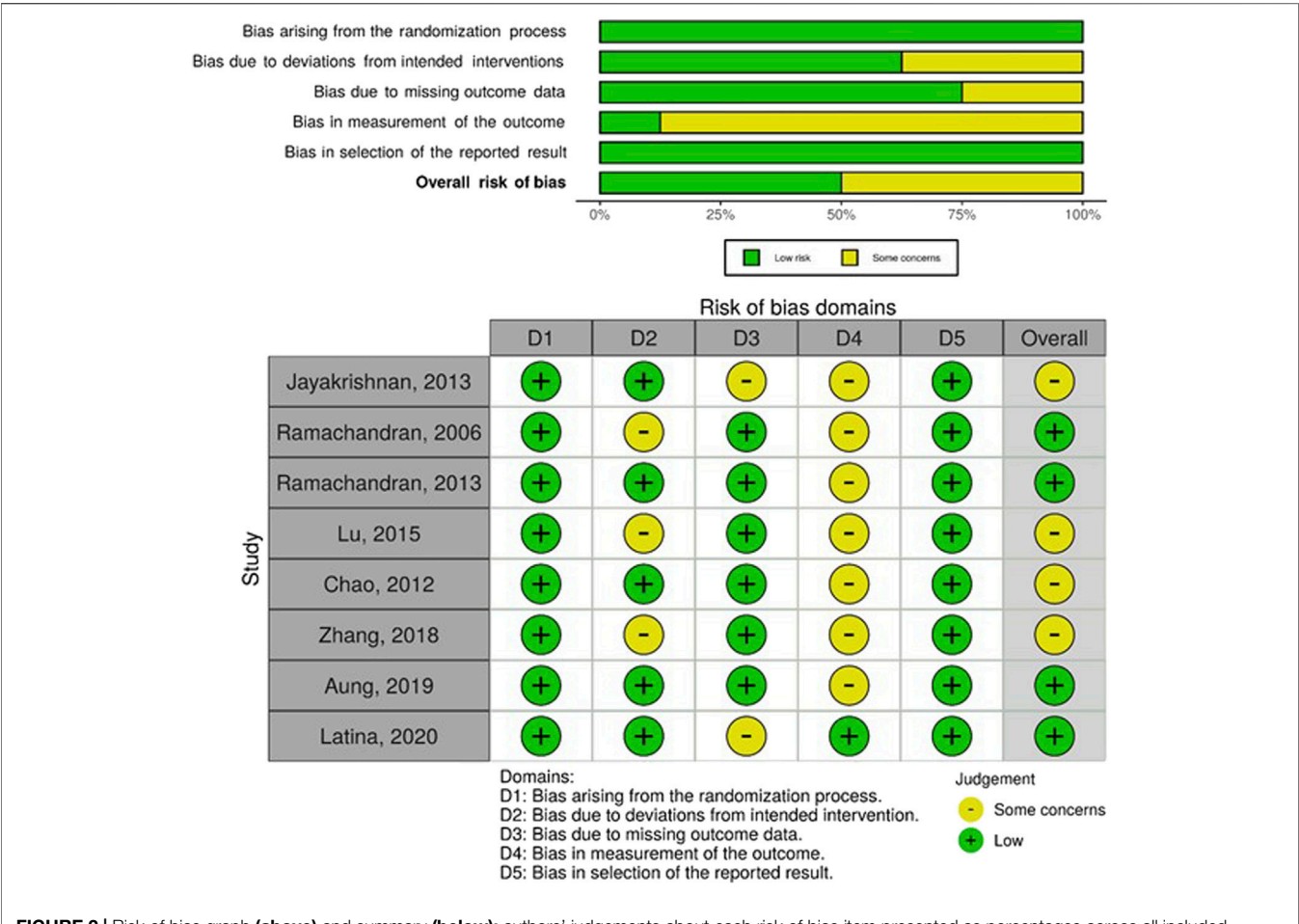

**FIGURE 2 |** Risk of bias graph **(above)** and summary **(below);** authors' judgements about each risk of bias item presented as percentages across all included randomized controlled trials, Community-based interventions for cardiovascular disease prevention in low- and middle-income countries: a systematic review, low- and middle-income countries, 2000–2019.

resource persons [26, 30, 38, 47, 55], and researchers and experts [26, 33–35, 37, 46, 48, 49, 51]. The intervention settings were community [26, 30, 32–42, 45, 49, 51, 52, 55, 56], community health care facilities [26, 30, 47, 50], schools [26, 30, 47, 55], workplaces [47, 55], neighbourhoods [47], and churches and mosques [26, 30]. The interventions lasted between 6 months [35, 53] and 5 years [26] while the follow-up period ranged from 1 year [33, 35, 42, 51–53, 56] to 5 years [26]. In five of the studies, the control group received a mild intervention [33–35, 37, 51] while for the rest, it was usual care or no intervention (**Table 1**).

## Effectiveness of Community-Based Interventions for CVD Prevention

**Table 2** summarizes the effect of the intervention on the behavioral and metabolic outcomes.

## Primary Outcomes
### Knowledge on CVDs and Risk Factors

Six studies half of which had moderate/some concerns RoB [40, 46, 47] and the rest had a high/serious RoB [38, 45, 49] examined

changes in CVD knowledge following implementation of community-based interventions. In five of the studies, knowledge significantly improved in the intervention groups related to dietary and lifestyle factors [38, 47, 49], hypertension [38, 46, 49] and diabetes [40] compared to the control groups. In another study in China, although tobacco related knowledge increased, diet and physical activity knowledge decreased in the intervention compared to the control group [47]. In a study with a high RoB carried out in India, there was no statistically significant effect of the intervention on knowledge about the six lifestyle factors affecting CVD risk [45]. Interventions that were effective in enhancing CVD knowledge majorly involved training, community mobilization, health education and consultation delivered through campaigns, group meetings, workshops, use of mobile technologies and of health workers, community health workers or peers.

## Physical Activity

A total of 23 studies recorded changes in physical activity across study populations. Among these, 16 studies compared improvements in physical activity between the intervention

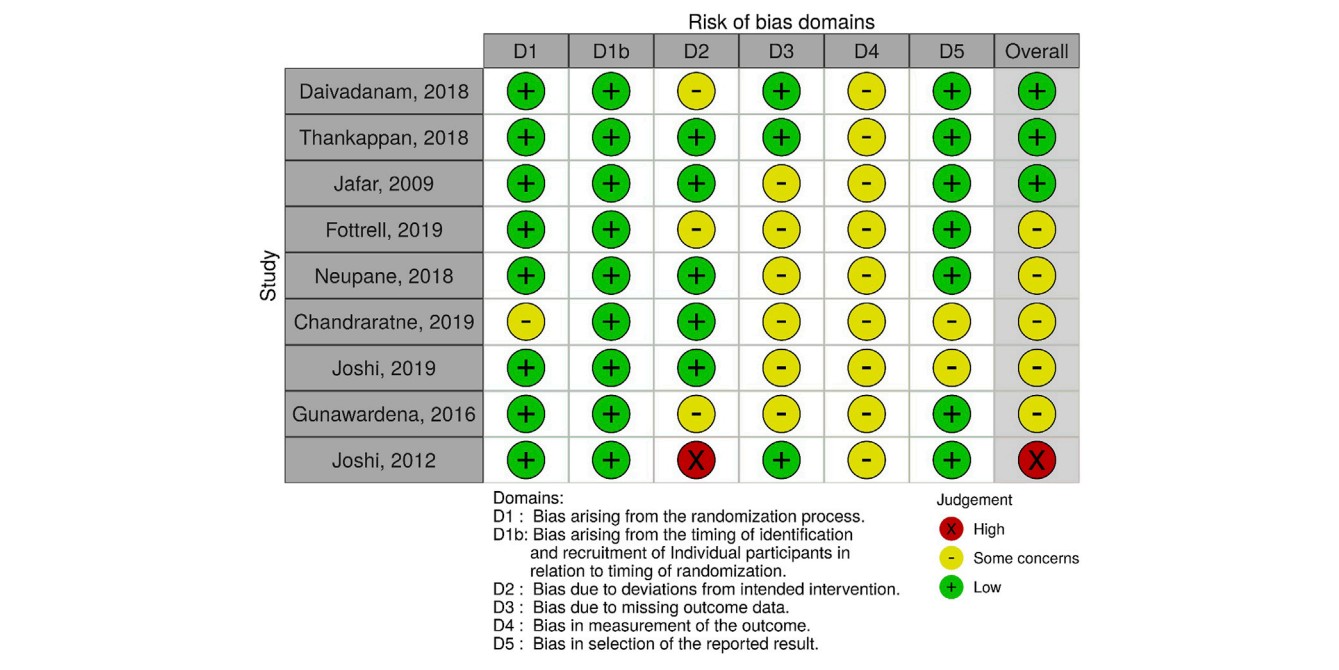

**FIGURE 3 |** Risk of bias summary; authors' judgements about each risk of bias item for each included cluster randomized controlled trial, Community-based interventions for cardiovascular disease prevention in low- and middle-income countries: a systematic review, low- and middle-income countries, 2000–2019.

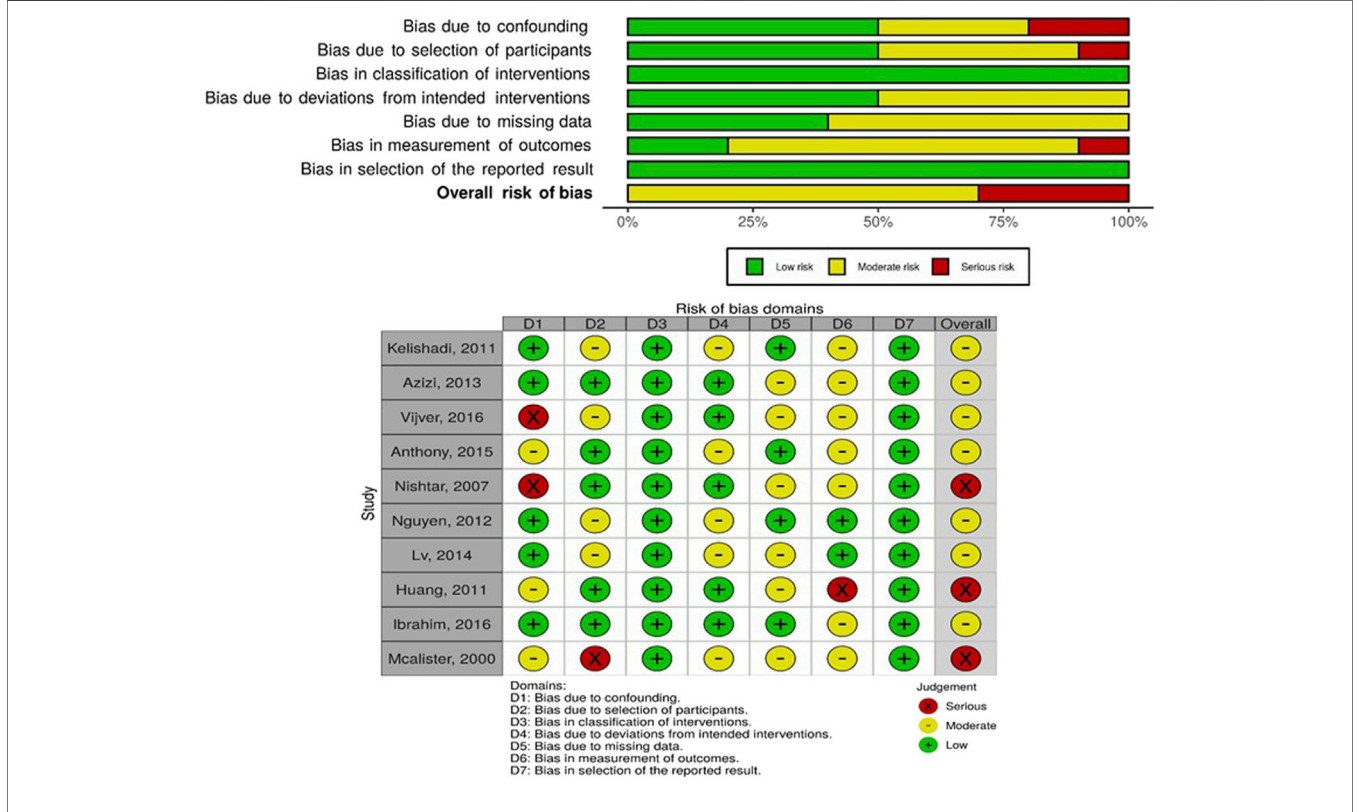

**FIGURE 4 |** Risk of bias graph **(above)** and summary **(below)**; authors' judgements about each risk of bias item presented as percentages across all included controlled before and after studies, Community-based interventions for cardiovascular disease prevention in low- and middle-income countries: a systematic review, low- and middle-income countries, 2000–2019.

**TABLE 1 |** Characteristics of included studies and community-based interventions, Community-based interventions for cardiovascular disease prevention in low- and middle-income countries: a systematic review, 2000–2019.

| Author | Country (income category) | Study design | Study aim | Context | Target population | Age | Participants | Intervention | Length | Provider | Setting | Follow-up time | Control |
|---|---|---|---|---|---|---|---|---|---|---|---|---|---|
| Baghaei, 2010 [25] | Iran (upperMIC) | Controlled before and after | To demonstrate the efficacy of the isfahan healthy heart program interventional strategies to improve lifestyle behaviors in a population at risk for developing cardiovascular diseases | Urban and rural isafayan (intervention) and najafabad (reference) | Adults with atleast one CVD risk factors including high blood pressure, diabetes mellitus, dyslipidaemia, metabollic syndrome, obesity and smoking | 19 years and above | 9,411 | 1) policy/Environmental strategies. 2) community outreach/Program services and. 3) surveillance | 5 years | Mass media, campaigns | Community | 5 years | No intervention |
| Kelishadi, 2011 [26] | Iran (upperMIC) | Controlled before and after | To investigate the effect of a comprehensive community trial on behavioral modification after 2 years of intervention | Urban and rural areas (two cities for intervention and one for control) | Adults in selected cities | 19 years and above | 12,514 | Community mobilization through training the trainers, activities to improve knowledge, attitude and practice, sport and physical activity, education through social gatherings, and involvement of community leaders | 2 years | Mass media, health professionals and market leaders, key nongovernmental organization staff, and local political decision makers (county, municipal, and provincial leaders) | Community | 2 years | No intervention |
| Rabiei, 2010 [27] | Iran (upperMIC) | Controlled before and after | To present the changes in PA habits after 2 years of intervention for increasing PA. | Urban and rural areas of isfahan and najafabad | Adults in selected cities | 19 years and above | 12,514 | Educational interventions–face to face or through class or education materials and camapigns. Environmental interventions–infrastructures for cycling, walking and use of public transportation. Legislative interventions–rules around physical activity in schools and directives on exercise time | 2 years | Mass media, health professionals and market leaders, key nongovernmental organization staff, and local political decision makers (county, municipal, and provincial leaders) | Community | 2 years | No intervention |
| Sarrafzadegan, 2009 [28] | Iran (upperMIC) | Controlled before and after | To study the feasibility and impact of a comprehensive, integrated, community-based program directed toward reducing modifiable risk factors for cardiovascular disease | Urban and rural areas | General population, the high-risk groups as well as specific target groups in urban and rural areas of the intervention. communities (isfahan and najaf-abad) | 19 years and above | 12,514 | Radio and television series for a healthy diet, exercise, educational cartoons and films for children, etc., different smoking cessation campaigns like the international smoking cessation program "quit andWin" or big campaigns about passive smoking, healthy lifestyle | 5 years | Multiple including mass media, community health professionals, role models and opinion leaders, religious leaders etc. | Community | 5 years | No intervention |
| Sarrafzadegan, 2009 [29] | Iran (upperMIC) | Controlled before and after | To assess the effects of a comprehensive, integrated community-based lifestyle intervention on diet, physical activity and smoking in two iranian communities | Urban and rural areas | General population as well as specific groups within the intervention communities | 19 years and above | 12,514 | Public education through the mass media, inter-sectoral cooperation and collaboration, community participation, education and involvement of health professionals, marketing and organizational development, legislation and policy development or enforcement, and research and evaluation | 4 years | Multiple | Community and health facility | 4 years | No intervention |

**TABLE 1 |** (*Continued*) Characteristics of included studies and community-based interventions, Community-based interventions for cardiovascular disease prevention in low- and middle-income countries: a systematic review, 2000–2019.

| Author | Country (income category) | Study design | Study aim | Context | Target population | Age | Participants | Intervention | Length | Provider | Setting | Follow-up time | Control |
|---|---|---|---|---|---|---|---|---|---|---|---|---|---|
| Azizi, 2013 [30] | Iran (upperMIC) | Controlled before and after | To assess the effects of lifestyle modifications on metabolic syndrome and some of its components in urban population of the lipids and glucose study | Urban residents of district 13 within tehran, the capital of Iran | Individuals who had returned for follow up after 3.6 years | 20–74 years | 6,870 | Nutrition education classes; healthy nutrition messages written in health newsletter. Distribution of pamphlets, brochures and booklets written on smoking, nutrition, physical activity and coping with stress. Nutrition intervention in nutrition clinics for subjects with diseases such as diabetes, overweight and obesity, dyslipidemia and hypertension | 3.6 years | Health volunteers and clinic staff | Community, school and facility based | 3.6 years | Not clear |
| Mirmiran, 2008 [31] | Iran (upperMIC) | Controlled before and after | To determine the effectiveness of nutrition intervention on non-communicable disease risk factors among tehranian urban adults | Residents in an urban distrct of Iran | Subjects aged 3 years and above | 18–74 years | 578 | Nutrition intervention introduced for all individuals aged 3 years and over in health care centers, schools and public places. At health facilities, family members invited and trained with face to face contact between educators and participants. In schools education by teachers, cooperation societies and group based activities including fairs and competitions. Foods at school canteen changed according to guidleines | 3.8 years | Educators at health facilities and trained teachers, parent-teacher cooperation societies, and group-based activities such as fairs and competitions in schools | Community, health care centers, schools, and public places | 3.8 years | No nutrition interventions |
| Vijver, 2016 [32] | Kenya (lower MIC) | Controlled before and after | To evaluate the impact of a community-based CVD prevention intervention on blood pressure (BP) and other CVD risk factors in a slum setting in nairobi, Kenya | Urban poor living in two slums, korogocho and viwandani | Adults living in the slums | 35 years and above | 2,764 | Raising awareness through mass media and public places, improving access to screening through BP measurement by community health workers, facilitating access to treatment through providing vouchers, and promoting long-term retention in care through encouraging visits, patient support groups and providing subsidies | 18 months | Radio, CHWs, health workers | Community | 18 months | Access to CVD standard of care (access outside the slum) |
| Daivadanam, 2018 [33] | India (lower MIC) | Cluster randomized controlled trial | To test the effectiveness of a sequential stage-matched intervention strategy to increase the daily intake of fruits and vegetables by an absolute 20% from baseline in the intervention arm over a one-year intervention period | Rural India | Adult males or females in the geographical area | 25–45 years | 479 | Initial face-to-face counseling, home-visits, general awareness sessions for community members in groups. Sequential stage-matching to differentiate households based on readiness to change dietary behavior in order to determine the strategies to be delivered to each household | 9 months | Counselors and psychology students trained in brief intervention. Community volunteers trained to deliver specific strategies and study coordinator | Community | 12 months | Information on recommended levels of intake of the five dietary components, and general dietary information leaflets |

**TABLE 1** | (*Continued*) Characteristics of included studies and community-based interventions, Community-based interventions for cardiovascular disease prevention in low- and middle-income countries: a systematic review, 2000–2019.

| Author | Country (income category) | Study design | Study aim | Context | Target population | Age | Participants | Intervention | Length | Provider | Setting | Follow-up time | Control |
|---|---|---|---|---|---|---|---|---|---|---|---|---|---|
| Thankappan, 2018 [34] | India (lower MIC) | Cluster randomized trial | To evaluate the effectiveness of a peer-support lifestyle intervention in preventing type 2 diabetes among high-risk individuals identified on the basis of a simple diabetes risk score | Neighboorhoods in close proximity | Individuals with an indian diabetes risk score ≥60 and free from diabetes on an oral glucose tolerance test | 30–60 years | 1,007 | Community-based peer-support program, a range of group session and community activities to support lifestyle change on diabetes and risk factors, nutiriton and physical activity held in community facilities such as schools and community halls and peer group sessions were held monthly except for the first two done fortnightly | 12 months | Diabetes and lifestyle experts and trained peer leaders | Community | 24 months | An education booklet with lifestyle change advice |
| Jayakrishnan, 2013 [35] | India (lower MIC) | Randomized controlled trial | To assess the effectiveness of a cessation intervention in rural Kerala state, India | 4 randomly allocated. Community development blocks in rural. Thiruvananthapuram district of Kerala state in south India | Current daily smoking resident males | 18–60 years | 928 | Awareness on tobacco hazards, anti-tobacco leaflets, a "how to quit tobacco" guide and a quick reference guide titled "how to quit tobacco". Group counseling and face-to-face individual counseling for participants through house visits or mobile phones | 6 months | Medical social workers | Community | 12 months | General awareness training on tobacco hazards along with an anti-tobacco leaflet. |
| Ramachandran, 2006 [36] | India (lower MIC) | Randomized, controlled trial | To determine whether the incidence of type 2 diabetes could be modified by interventions | An urban population in India | Middle-class working indian men with impaired glucose tolerance | 35–55 years | 502 | Lifestyle modification group including advice on healthy diet and regular physical activity. Metformin (MET) group with diaries to record daily consumption of tablets. LSM + met group–subjects were given LSM plus MET. All intervention groups received monthly phone calls for continued motivation | 30 and 36 months | Health workers | Community | 30 and 36 months | Standard health care advice |
| Ramachandran, 2013 [37] | India (lower MIC) | Randomized controlled trial | To assess whether mobile phone messaging that encouraged lifestyle change could reduce incident type 2 diabetes in indian asian men with impaired glucose tolerance | An urban population in India | Working indian men with impaired glucose tolerance | 35–55 years | 537 | Personalized education and motivation about healthy lifestyle principles and frequent mobile phone messages–with information about healthy lifestyle, the benefits of physical activity and diet, cues to start, and strategies to avoid relapse | 2 years | Study staff, mobile phone delivery website | Community | 3 years | Standard lifestyle modification advice at baseline only |
| Nishtar, 2007 [38] | Pakistan (lower MIC) | Controlled before and after | To examine changes in knowledge and CVD risk factors among the community following an intervention | 85.5% of district is rural whereas 14.5% is urban with high rates of poverty, illiteracy, low education levels and high unemployment | Males and females resident in the district | 18–65 years | 604 | Community health education during meetings through organized groups, mass media interventions, training of health professionals through one-day workshops and lady health workers, and health education through lady health workers who undergo 3 days monthly training | 2 years | Trained officer for health education, radio, lady health workers | Community | 2 years | No intervention |

**TABLE 1 |** (*Continued*) Characteristics of included studies and community-based interventions, Community-based interventions for cardiovascular disease prevention in low- and middle-income countries: a systematic review, 2000–2019.

| Author | Country (income category) | Study design | Study aim | Context | Target population | Age | Participants | Intervention | Length | Provider | Setting | Follow-up time | Control |
|---|---|---|---|---|---|---|---|---|---|---|---|---|---|
| Jafar, 2009 [39] | Pakistan (lower MIC) | Cluster randomized controlled trial | To assess the effectiveness of two community-based interventions on blood pressure in hypertensive adults | Households in karachi, the most populous city in Pakistan (mean household monthly income, $70) | Residents in the area and had known hypertension or consistently elevated blood pressure on 2 separate visits | 40 years and above | 1,341 | Family based home health education from lay health workers and annual training of general practitioners in hypertension management. HHE alone: Trained CHWs deliver behavior changing communication strategies to convey standardized health education messages to households. GP alone: training of at least two thirds of the GPs in area. HHE and GP combined | 2 years | Lay health workers and GPs | Community | 2 years | No intervention |
| Fottrell, 2019 [40] | Bangladesh (lower MIC) | Cluster randomized trial | To assess the effects of mHealth and community mobilization on the prevalence of intermediate hyperglycaemia and diabetes among the general adult population | Rural Bangladesh | Men and non-pregnant women | 30 years and above | 12,280 for cross sectional and 2,470 for cohort | Participatory community mobilization–involved 18 monthly group meetings, led by lay facilitators, applying a participatory learning and action (PLA) cycle focused on diabetes prevention and control. mHealth mobile phone messaging–twice-weekly voice messages over 14 months promoting behavior change to reduce diabetes risk | Mhealth–14 months. Community mobilization 18 months | Lay facilitators for community mobilization. Mobile phones for voice messages | Community | 18 months and 2 years | Usual care |
| Nguyen, 2012 [41] | Vietnam (lower MIC) | Controlled before and after | To evaluate the impact of healthy lifestyle promotion campaigns on CVD risk factors in the general population in the context of a community-based program on hypertension management | Rural population | Community inhabitants | 25 years and above | 4,650 | A hypertensive-targeted management program (monthly BP check, training health workers and multidrug therapy) integrated with a community-targeted health promotion on smoking cessation, reducing alcohol consumption, encouraging physical activity and reducing salty diets (healthy individuals) delivered through lifestyle promotion campaign through broadcasting, leaflets or meetings | 3 years | Health workers, mass media | Community | 3 years | Routine conventional healthcare services |
| Chandraratne, 2019 [42] | Sri Lanka (lower MIC) | Cluster randomized trial | To investigate the effect of an intervention with youth on cardiovascular disease risk factors of community adults | A semi-urban area | Adults within selected divisions | 30–59 years | 512 | Youths as change agents taking body weight and blood pressure measurements and proposed healthy lifestyle behaviors to adults | 12 months | Trained youths who are members of a club | Community | 12 months | No intervention |

**TABLE 1** | (*Continued*) Characteristics of included studies and community-based interventions, Community-based interventions for cardiovascular disease prevention in low- and middle-income countries: a systematic review, 2000–2019.

| Author | Country (income category) | Study design | Study aim | Context | Target population | Age | Participants | Intervention | Length | Provider | Setting | Follow-up time | Control |
|---|---|---|---|---|---|---|---|---|---|---|---|---|---|
| Joshi, 2019 [43] | India (lower MIC) | Cluster randomized trial | To assess impact of CHW based interventions in reducing CVD risk factors in rural households in India | Rural regions in India | All households with individuals of age ≥35 years living in these villages | Individuals aged 35 years and above | Individuals resident within selected villages | CHWs delivered risk-reduction advice and monitored risk factors during 6 household visits. CHWs measured blood pressure, ascertained and reinforced adherence to prescribed therapies. The CHWs also placed short goal-directed slogans printed on common household objects in the household to promote integration of preventive therapies with activities of daily living | 12 months | Community health workers | Community | 12 and 18 months | Did not receive CHW visits but had access to a clinic which was run in the sub-center location |
| Gunawardena, 2016 [44] | Sri Lanka (lower MIC) | Cluster randomized controlled trial | To examine the effect of the child-initiated intervention on weight, physical activity and dietary habit of their mothers | Semi-urban area of colombo | Sri lankan mothers with a school-aged child | Mean age between 37.5 and 38.5 years | 308 | Trained health promotion facilitators visited the intervention schools and delivered the intervention in the form of a series of discussion on health and well-being with the selected students with emphasis on their mothers | 12 months | Students | Community | 12 months | No intervention |
| Joshi, 2012 [45] | India (lower MIC) | Cluster randomized controlled trial | To develop, implement, and evaluate 2 CVD prevention strategies that could. Potentially be delivered by NPHWs: 1 based on a clinical. Approach and 1 based on health promotion | Rural villages in the east and west godavari districts of Andhra Pradesh in India | Residents of the study area | 30 years or older | 3,711 | The campaign included posters, street theater, rallies, and community presentations designed to convey messages about stopping tobacco use, heart-healthy eating, and physical activity | 18 months | Posters, street theater, rallies | Community | 18 months | In the villages assigned to the control group, there was no additional health promotion campaign planned |
| Lu, 2015 [46] | China (upper MIC) | Randomized controlled trial | To evaluate community-based health education strategies in the management of hypertensive patients with low socioeconomic status in dongguan city, China | Community health service center in liabout town in China | Hypertensive patients managed at a community health service center | 40–75 years | 360 | Three health education groups. Self-learning reading group - orientation on reading materials to learn about hypertension through posters and booklets. Monthly regular didactic lecture group by phone invitation with lecture lasting about 30 min. Monthly interactive education workshop–active involvement of participants in use of visual health education tools | 2 years | Health education materials developed by CVD experts | Community | 2 years | Groups controlled for each other |

**TABLE 1 |** (*Continued*) Characteristics of included studies and community-based interventions, Community-based interventions for cardiovascular disease prevention in low- and middle-income countries: a systematic review, 2000–2019.

| Author | Country (income category) | Study design | Study aim | Context | Target population | Age | Participants | Intervention | Length | Provider | Setting | Follow-up time | Control |
|---|---|---|---|---|---|---|---|---|---|---|---|---|---|
| Lv, 2014 [47] | China (upper MIC) | Controlled before and after | To assess the short-term impact of a comprehensive, community-based multilevel intervention on knowledge, beliefs and practices with respect to smoking, physical activity and diet in hangzhou, China | Three adjacent districts located in a central geographical location of hangzhou were included | Individuals who had lived in the local district for at least 1 year | 18–64 years | 2016 | Community mobilization, structural change in individual, social, physical and policy environment, health education and social marketing | 2 years | Staff of local organization, health workers, community public health assistants | Neighboorhoods, schools, workplaces and community health center | 2 years | Routine NCD prevention and control practices |
| Chao, 2012 [48] | China (upper MIC) | Randomized controlled trial | To evaluate the impact of community-based health management on the health of the elderly through an RCT in nanjing, China | Nanjing in southeastern China is the provincial capital Of jiangsu province–one of the most developed provinces in China | Elderly persons resident in the area | 60 years and above | 2,400 | Health management program including 1) health record establishment; 2) health evaluation; and 3) health management, including diet advice, psychological aspects of health, a tailor-made exercise program, education/skills training on health self-management, telephone consultation, lectures on health, and distribution of health promoting materials | 18 months | Specifically trained community health service center staff, managers and related researchers | Community health service center | 18 months | Usual care |
| Huang, 2011 [49] | China (upper MIC) | Controlled before and after | To evaluate the effects of a community intervention program, which focused on improving the hypertension knowledge, diets and lifestyles in a rural Chinese area | Rural area with highlands and agriculture resources (oranges and tea) | Rural residents from xiaoxita and fenxiang towns | 35 and above years | 1,632 | Training local staff on guidelines for hypertensions prevention and treatment, and education of participants through using hypertension education and dietary and lifestyle guidance. Provided pamphlets with information on dietary and lifestyle behaviors and one salt spoon (for 2 g) to households | 3 years | Trained local healthcare staff, media, research team and materials | Community | 3 years | Access to normal, standard health care during the study period |
| Zhang, 2018 [50] | China (upper MIC) | Randomized controlled trial | To investigate the effectiveness of the older-centered integrated health management model project for multiple lifestyle behaviors in the elderly | Community health centers in nanjing, China | Elderly persons who had lived in the community for atleast 2 years | 60–80 years | 671 | One hour health education program every two months covering several topics including chronic disease risk factors, psychosocial support and lifestyle changes, and a two year health management program that provided them with information, skills and tools for self-management, family and community management | 2 years | Community health service center staff | Hospital or community center | 24 months | Usual care |

**TABLE 1 |** (*Continued*) Characteristics of included studies and community-based interventions, Community-based interventions for cardiovascular disease prevention in low- and middle-income countries: a systematic review, 2000–2019.

| Author | Country (income category) | Study design | Study aim | Context | Target population | Age | Participants | Intervention | Length | Provider | Setting | Follow-up time | Control |
|---|---|---|---|---|---|---|---|---|---|---|---|---|---|
| Ibrahim, 2016 [51] | Malaysia (upper MIC) | Controlled before and after | To determine the effects of a community-based lifestyle intervention delivered to adults with pre-diabetes and their health-related quality of life as compared to the usual care group | Sub-urban communities in seremban. Malaysia | Participants with prediabetes | 18–65 years | 297 | Twelve group-based sessions of 90 min each and minimum of two individual counseling sessions with the dietician and researcher to reinforce behavioral change. Group sessions (i.e., lecture, seminar, group work or discussion). Community volunteers trained in a two-day training workshop prior to the delivery of intervention | 12 months | Dietetician and researcher | Community | 12 months | Standard health education from primary care providers in the clinic and pamphlets. And booklets about various health topics |
| Mcalister, 2000 [52] | Russia (upper MIC) | Controlled before and after | To test the feasibility and effectiveness of smoking reduction that have proven to be useful in Finnish karelia | Impoverished area with difficult economic conditions, low socio-economic status, poorly funded health system and high mortality | Daily smokers in pitkäranta and adjacent district of suojärvi | 25–64 years | 378 | A smoking cessation campaign "quit and win" with media publicity (role modeling) and community support | 1 year | Staff of the pitkäranta central hospital | Community | 12 months | No intervention |
| Aung, 2019 [53] | Thailand (upper MIC) | Randomized controlled trial | To compare levels of smoking cessation between the intervention new service package arm and the control routine service arm over a six-month period and also compare the smokerlyzer-confirmed smoking cessation rates | Rural districts where people often grow tobacco in their gardens and consume home-made hand-rolled cigarettes | Patients with hypertension and diabetes in northern Thailand | 35–80 years | 319 | Regular patient motivation by the same nurse over a 3 month period, a monthly piCO + smokerlyzer test for 3 months, continual assistance from a trained family member, using a smoking-cessation- diary; and optional nicotine replacement chewing gum therapy | 6 months | Health workers and family member | Starts at primary care into community | 12 months | Routine service comprising of brief counseling and casual follow-up |
| Latina, 2020 [54] | Grenada (upper MIC) | Randomized controlled trial | To test the effectiveness of peer support strategy for cardiovascular risk reduction | Residents were from five parishes around the island | Adults from the Grenada heart project cohort study of age with at least two CV risk factors | 18–85 years | 402 | CHWs were trained to deliver the community-based program over a 5 days course. The intervention group was organized into groups of 8–12 individuals in their local parish. A "peer leader" underwent additional training in leadership and communication and health behavior communication | 12 months | Peer leaders | Community | 12 months | Series of educational lectures at the time of enrollment, followed by self-management for 1 year |
| Anthony, 2015 [55] | China (UpperMIC), India (lower MIC), Mexico (upperMIC) | Controlled before and after | To reduce risk factors in workplace settings in low- and middle-income countries | Intervention and control areas in each country had similar demographic. And socioeconomic characteristics with a population size. Between 150,000 and 200,000 | All workers aged 18–64 years in each workplace | 18–64 years | 12,136 | Health education, structural change though policies, mandates and restrictions, and community mobilization including no smoking days and smoking bans | 18–24 months | Community coalitions consisting of key decision makers, media | Workplaces, schools, and community | 18 and 24 months | No intervention |

**TABLE 1** | (Continued) Characteristics of included studies and community-based interventions, Community-based interventions for cardiovascular disease prevention in low- and middle-income countries: a systematic review, 2000–2019.

| Author | Country (income category) | Study design | Study aim | Context | Target population | Age | Participants | Intervention | Length | Provider | Setting | Follow-up time | Control |
|---|---|---|---|---|---|---|---|---|---|---|---|---|---|
| Neupane, 2018 [56] | Nepal (low) | Cluster randomized trial | To investigate the effectiveness of a population-level intervention led by existing community health workers in reducing the burden of hypertension in a low-income population | community-based survey in lekhnath municipality, Nepal | Adults within area | 25–65 years | 1,638 | Female CHWs underwent 5 days training on BP measurement. Lifestyle intervention led by female community health volunteers (home visits every 4 months for lifestyle counseling and blood pressure monitoring). FCHVs visited selected households three times a year (every 4 months) to provide health promotion counseling and to measure blood pressure | 12 months | Female CHWs | Community | 12 months | Usual care |

and control groups and eight with low (1), moderate/some concerns (7) RoB found changes in favor of the intervention group in engaging in physical activity [26, 41, 44], leisure-time activity [44], proportion of participants physically active [30, 32, 51], adherence to physical activity [50] and energy expenditure for physical activities [26, 39]. In one study with a serious RoB, significant improvements were higher in the control group [49] while there was no significant difference between groups in the remaining studies all of which were of low, moderate or some concerns RoB [34, 37, 42, 43, 54–56]. In seven studies, differences in physical activity were tested within groups and significant improvements were noted in the intervention group [26, 27, 47, 48], both intervention and control groups [46] or none of them [36, 38, 40]. Some studies compared physical activity levels between males and females and found more activity among males than females [41] and higher activity levels among females compared to males [26]. Majority of effective studies utilized health education through mass media, public places, and information, education and communication materials, training of health workers and community volunteers, community mobilization through campaigns and structural changes in the environment.

## Diet

Twenty-one studies determined changes in diet following implementation of interventions. Fruits and vegetable consumption significantly improved among intervention compared to control group in five studies of low [34], some concerns [40], moderate [26, 55], or high RoB [38] unlike others [32, 33, 44, 45, 56]. In one study with some concerns RoB, the changes were only observed in fruit intake [42]. Several studies also showed significant differences in favor of the intervention in lower salt consumption [41, 49, 55], fat intake [26, 45, 49] and snacks consumption [42] when compared with control groups. Other studies measured overall diet quality and found significantly greater improvements in diet score or adherence to recommended diets in intervention populations relative to control group [37, 48, 50]. In six studies, there were no significant differences between the intervention and control groups in dietary components examined [30, 36, 46, 47, 51, 56]. Effective dietary interventions mostly focused on providing advice on healthy diets, community health education during meetings and public places, community mobilization, tailoring interventions based on readiness to change and follow up of persons such as through phone calls.

## Alcohol Use

Eight studies explored outcomes related to alcohol consumption as a risk factor for CVD [32, 34, 41, 42, 46, 49, 50, 56]. In two studies of low and serious RoB, intervention group participants exhibited a greater reduction in alcohol consumption compared to control participants [34, 49] while in another study of moderate RoB, significant differences were observed in both intervention and control communities [41]. Effective interventions for alcohol reduction were mainly community-based peer support with group sessions, community-wide activities such in public places, health education through

pamphlets, newspapers, classes and training of health workers. In the remaining studies with some concerns RoB, there was no significant reduction in proportion of persons consuming alcohol [42, 46, 56] or adhering to its moderate use [50]. On the other hand, significant reduction in alcohol use at population level and among patients with hypertension was observed in the control group of a study with moderate RoB conducted in Kenya [32].

## Smoking

Among the 19 studies that examined changes in smoking levels, nine with low (2), moderate/some concerns (4), high/serious (3) RoB found a statistically significant difference in smoking prevalence in favor of the intervention compared to the control group [26, 35, 38, 39, 43, 47, 52, 53, 55]. Indeed, these differences were in smoking cessation [26, 35, 39, 47, 52, 53, 55] and reduced use of smokeless tobacco [38, 43]. In two controlled before and after studies with moderate RoB, stratified analysis found significantly greater reductions in smoking cessation among men but not women [26, 55]. Effective smoking interventions involved health education through social gatherings, pamphlets, brochures and booklets; community mobilization through training, involvement of local leaders and use of media campaigns; and group or individual counseling. In ten studies with low (3), moderate (2), some concerns (3), high (1) and serious (1) RoB, there was no statistically significant difference in smoking measures between the intervention and control groups [32, 34, 41, 42, 45, 46, 49, 50, 54, 56].

## Secondary Outcomes
### Body Weight

Fourteen studies assessed changes in body weight among study participants with six of low (2), moderate (1) and some concerns (3) RoB reporting significant reduction in body mass index [42, 44, 46], waist hip ratio [48], body weight [30, 36, 42, 44] and abdominal obesity [30] in intervention compared to control groups. Eight studies of low (3), some concerns/moderate (4) and high (1) RoB did not find any significant difference between groups [34, 37, 39, 41, 45, 50, 51, 55]. Health education groups, health management programmes including training and counseling, nutrition education classes, training and engaging community volunteers were effective interventions for reducing body weight indicators.

## Blood Pressure

Following implementation of interventions among the population groups, compared with the control groups, in seven of the 16 studies with low (1) and some concerns/ moderate (6) RoB, the intervention groups registered significant reductions in systolic [39, 48, 56] and diastolic blood pressure [30, 51] or both [41, 50]. In addition, two studies with high/serious RoB demonstrated a positive effect of the intervention on awareness, treatment and control of hypertension [49] and adherence to antihypertensive drugs [43]. Studies that showed effectiveness in blood pressure outcomes involved several strategies including health education awareness raising through mass media and public

places, trainings, lifestyle promotion campaigns, health management programmes providing information, skills and tools for self-management, counseling sessions and improving access to blood pressure screening. There was no significant difference observed between intervention and control groups in systolic [32, 43, 51] and diastolic [39, 48] blood pressure or both [34, 37, 40, 42, 45], and hypertension prevalence [30, 32, 40] and control [39, 40, 46] even though within group differences were observed in some of the studies.

## Blood Glucose

The ten studies that evaluated blood glucose related outcomes with low (4) and some concerns/moderate (6) RoB found significant reductions in fasting blood sugar [30, 48, 50, 51, 54], and prevalence of diabetes and intermediate hyperglycaemia [40], risk of developing diabetes [34] and incidence of diabetes [26, 36, 37] in the intervention compared to control group. In one study with a low RoB, there was no significant difference in the incidence of diabetes between the intervention and control groups [34]. In another study with some concerns RoB, although differences were observed in the community mobilization arm, the m-health intervention arm did not influence the combined prevalence of intermediate hyperglycaemia and diabetes or the incidence of diabetes [40]. Interventions among effective studies included health education and advice, trainings, nutrition education through classes and print media such as brochures and pamphlets, provision of information and tools for self-management, secondary prevention activities in clinics and counseling.

## Lipids

Seven studies evaluated changes in blood lipids and in four studies with low (1) and some concerns/moderate (3) RoB, significantly lower measures were registered for low density lipoprotein (LDL) [30, 37], triglycerides [30, 46] and total cholesterol [30, 46, 50] and a high prevalence of high density lipoprotein (HDL) [51] in intervention compared to control groups. Conversely, within the intervention group of a study in Iran with moderate RoB, prevalence of low HDL cholesterol increased at follow-up [30]. Health education sometimes personalized for individuals to support lifestyle changes or conducted in groups through classes and print media or mobile messages, health management programmes, and secondary prevention activities in clinics formed effective interventions. There was no statistically significant difference between intervention and control groups in total cholesterol [34, 46], LDL cholesterol [34, 46], serum cholesterol [37] and triglycerides [34, 37, 48] in other studies.

## DISCUSSION

This systematic review examined community-based interventions for CVD prevention in LMICs and their effectiveness. Among the 27 studies that were included in the review, most employed health education and awareness creation, trainings, lifestyle consultation and counseling and community

mobilization. Community-based interventions successfully improved population knowledge on CVD and risk factors and influenced physical activity and dietary practices unlike reduction in smoking and alcohol consumption. The interventions also led to significant improvements in blood pressure and blood sugar measurements. Overall, the effective interventions mostly involved community mobilization and social activities; health education and communication through use of information, education and communication materials including mass media; individual or group counseling; and trainings of providers including community health workers, peers or health workers.

Health education and health communication includes verbal and written measures to influence and empower individuals, populations, and communities to make healthier choices [58]. Many studies included more than one intervention strategy similar to other reviews [59] which is desirable as multi-strategy interventions have been demonstrated to be more effective in influencing uptake of behaviors compared to individual strategies [58]. These strategies aim at changing people's knowledge, attitudes and/or behaviors and are considered a crucial first step in addition to community mobilization activities that help create a favourable enabling environment [58]. In addition, with the huge knowledge and capacity gaps in CVD services delivery in many LMICs [60–62], the high number of training interventions is not surprising.

The review also examined the effectiveness of community-based interventions in increasing knowledge and uptake of CVD prevention practices. We found significant improvements in knowledge on behavioral and metabolic risk factors for CVD prevention in favor of the community-based interventions similar to a previous review [14]. These improvements were achieved mostly through training, community mobilization, health education and consultation [38, 40, 45, 46, 49]. Worthy of note is that knowledge on CVD is still very low in many LMICs [63, 64] and thus interventions are more likely to lead to significant changes. Regarding, changes in behavioral practices following implementation of intervention, greatest improvements were around physical activity with increase in proportion physically active and leisure-time activity. Also, improvements in dietary practices were observed with changes reported in fruit and vegetable consumption, salt intake, and fats and snacks intake. An earlier review also observed improvements in physical activity and diet as significant lifestyle changes reported by a sizable proportion of studies [19]. Improvements in other risk factors were inconsistent with only few studies reporting changes in smoking and alcohol consumption behaviors. In high-income contexts, changes in smoking have been reported following community programmes [59]. Changes in alcohol and smoking practices usually takes a longer time yet many studies in our review had shorter follow-up periods of one to two years. Moreover, sustaining changes in behavior over long periods has been shown to be challenging [65, 66]. In some studies, the intensity of the intervention was low with broad interventions and self-reported outcome measures which gaps should be bridged in future studies. Moreover, there is need for further studies of longer durations to examine changes in CVD

prevention practices and inform the growing evidence base for CVD prevention intervention effectiveness in LMICs. This is especially important because lifestyle interventions have been reported to be more cost effective than pharmacological interventions [67] which most LMICs cannot afford amidst the growing epidemic of NCDs [13]. When planning such intervention studies, multiple intervention strategies should be considered, adapted to context and informed by appropriate theories, frameworks and models and evaluated using robust study designs.

This review also examined changes in metabolic risk factors as secondary outcomes and found more evidence in favor of reduction in systolic blood pressure, diastolic blood pressure and blood glucose following community-based interventions which was from relatively lower ROB studies. Effectiveness of community-based intervention in reducing metabolic risk factors has been reported previously for several risk factors including systolic blood pressure [14, 19, 68, 69], diastolic blood pressure [69], incidence of diabetes [70] and HbA1C [70] including in high-income countries. Positive changes in behavioral risk factors should lead to improvements in metabolic risk factors for example changes in dietary behaviors and/or physical activity should impact blood pressure indicators. Although changes in metabolic risk factors were examined for only included studies as secondary outcomes and thus not comprehensive, this review provides insights into how community-based interventions impact both primary and secondary level CVD risk factors.

Overall, this systematic review notes that there have been interventions for CVD prevention in LMICs but these are mostly limited to middle-income Asian countries which could have been influenced by their higher CVD burden. The lack of studies from low-middle income sub-Saharan African countries is concerning and calls for systematic approaches to address the gaps. Targeted research funding and establishing dedicated research centers to support evidence generation and translation is recommended. Although majority of the studies were randomized controlled (17), only seven had a low risk of bias which affects the strength of currently available evidence. The design of future studies should consider measures to minimize identified sources of bias where possible including using objective measures of outcomes and/or using validated tools, properly designing interventions to avoid foreseeable deviations and applying appropriate techniques to deal with missing data. Moreover, randomized studies should ensure that randomization is effectively done to minimize baseline imbalances in study characteristics while non-randomized studies should control for most sources of confounding. We note that although the review intended to provide a comprehensive overview of community-based interventions and literature search widely done, the strict inclusion criteria designed to provide robust evidence on intervention effectiveness could have limited this. This review only considered studies published in the English language due to inadequate resources which could have led to publication bias hence future reviews should consider including other languages. We also excluded conference abstracts which

**TABLE 2 |** Effect of community-based interventions on study outcomes, Community-based interventions for cardiovascular disease prevention in low- and middle-income countries: a systematic review, 2000–2019.

| Author | Knowledge | Physical activity | Diet | Smoking | Alcohol | Body weight | Blood pressure | Blood glucose | Lipid profile |
|---|---|---|---|---|---|---|---|---|---|
| Baghaei, 2010 [25] | | Regular daily exercise significantly increased by about 45% among the high risk population in the interventional area (from 15 to 28%) | Improved fruit and vegetable consumption, more use of unsaturated fats and reduced salt intake among the high risk population in the intervention group compared to reference group ($p < 0.05$) | Smoking significantly decreased among the high risk persons in the intervention group compared to reference group ($p < 0.03$) | | | | | |
| Kelishadi, 2011 [26] | | Leisure time physical activity increased in women and declined in men of both communities with changes greater in the intervention area No significant change in transport physical activity in intervention area but sharply decreased in reference area Regular morning curricular exercise significantly was greater in the intervention community compared to control ($p < 0.001$) | Consumption of hydrogenated fat decreased significantly in the intervention community, but it remained nearly constant in the reference area ($p < 0.05$) Consumption of liquid oil (up to 2 times a day) increased in the intervention community, whereas it decreased in the reference area Consumption of salty/fat snacks slightly decreased in intervention area but increased in the reference area Fast food consumption did not significantly change in intervention area but increased sharply in reference area | Prevalence of current smoking decreased in men living in intervention area but increased slightly among those in reference area. However, no significant change was observed among women in both groups Attempt to smoking decreased among youths in intervention areas but remained constant in the reference area | | | | | |
| Rabiei, 2010 [27] | | From 2000 to 2002, the daily physical activity (PA) among both genders decreased in both intervention and reference communities Leisure-time PA increased significantly in the intervention area, but decreased in the reference area The transportation PA did not significantly change in the intervention area, but showed a remarkable decline in the reference area among both genders. No significant change in worksite PA. | | | | | | | |
| Sarrafzadegan, 2009 [28] | | Energy expenditure for total daily physical activities showed a decreasing trend in all areas, but the mean drop from baseline was significantly smaller in the intervention areas than in the control area (−68 metabolic equivalent task (MET) minutes per week vs. −114 MET minutes per week, respectively; $p < 0.05$). Leisure time devoted to physical activities showed an increasing trend in all areas | Changes from baseline in mean dietary score differed significantly between the intervention and control areas (+2.1 points vs. −1.2 points, respectively; $p < 0.01$), as did the change in the percentage of individuals following a healthy diet (+14.9% vs. −2.0%, respectively; $p < 0.001$) | Daily smoking had decreased by 0.9% in the intervention areas and by 2.6% in the control area at the end of the third year, but the difference was not significant. Analysis by gender revealed a significant decreasing trend in smoking among men ($p < 0.05$) but not among women | | | | | |

**TABLE 2 |** (*Continued*) Effect of community-based interventions on study outcomes, Community-based interventions for cardiovascular disease prevention in low- and middle-income countries: a systematic review, 2000–2019.

| Author | Knowledge | Physical activity | Diet | Smoking | Alcohol | Body weight | Blood pressure | Blood glucose | Lipid profile |
|---|---|---|---|---|---|---|---|---|---|
| Sarrafzadegan, 2009 [29] | | Intervention activities positively affected the total and leisure-time physical activities in men, but not women Total daily physical activity decreased in both groups over the years but trend was slower in intervention group (p < 0.001) | Intervention resulted into gradual improvement in the nutritional status compared to control with trends similar in men and women (p < 0.001) Dietary choice of both sexes showed modest degrees of improvement | Smoking status of men improved during the study period (except 2004), while the effect on women was not significant | | | | | |
| Azizi, 2013 [30] | | After intervention, chance for being less active was significantly higher in control men as compared to intervention men after 3.6 years: OR = 1.2 (1.01–1.44, p < 0.05) | No significant difference in energy intake and macronutrient consumption between two groups at baseline and after intervention | | | Prevalence of abdominal obesity increased significantly in both intervention and control groups. Comparison between groups revealed significant reduction in abdominal obesity in intervention group compared to control (OR 1.24, 1.07–1.44, p < 0.014) | Elevated blood pressure reduced in both intervention and control groups but difference between groups was not statistically significant | Prevalence of elevated fasting glucose increased only in the control group. On group comparisons, intervention group had significantly reduced elevated fasting glucose (OR 1.67, CI 1.43–1.95, p < 0.001) | Prevalence of low HDL cholesterol increased significantly in both groups while that of elevated triglycerides decreased significantly compared to baseline values. Compared to control, intervention significantly reduced elevated triglycerides (OR 1.18, CI 1.04–1.34, p < 0.014) and low HDL cholesterol (OR 1.52, CI 1.32–1.76, p < 0.001) |
| Mirmiran, 2008 [31] | | Mean dietary carbohydrate, mean dietary protein and fat intakes decreased in both group but significant in intervention group. After adjustments, only total dietary cholesterol had a significant decrease (p < 0.05. While the dietary vitamin a values decreased in controls, it increased in the intervention. Vitamin B6, B12, and C and zinc intakes increased significantly in both groups, while the iron intakes significantly decreased in both groups. No significant difference in energy and nutrient intakes between intervention and controls | | | Body mass index increased significantly in both groups but difference not statistically significant (p = 0.53) | Diastollic blood pressure decreased significantly in both groups but higher in the intervention group (p = 0.08). Systollic blood pressure increased significantly in the control group with no significant difference with intervention group (p = 0.13) | Significant decrease in fasting blood sugar in intervention group and significant increase in control group and difference between intervention and control group was statistically significant (p < 0.01) | Total cholesterol and HDL and LDL decreased significantly in both groups. There was significant decrease in total cholesterol in the intervention group compared to controls (p = 0.01) |
| Vijver, 2016 [32] | | Significant decrease in the numbers of those reporting inadequate physical activity among intervention compared control group at population level (OR 0.20, p < 0.001) | Insufficient intake of fruits and vegetables increased significantly at population level both in intervention (OR 1.30, 95% CI 1.08 to 1.56, p = 0.006) and control (OR 1.42, 95% CI 1.15 to 1.76, p = 0.001) settings but difference between groups was not significant (OR 0.88, 0.67 to 1.16, p = 0.375) | In the control group, there was significant decrease at population level in smoking (OR 0.73, 95% CI 0.56–0.95). The difference in the intervention group was not significant (OR 1.19, 95% CI 0.92 to 1.53, p = 0.181). Among patients with hypertension in the control group, smoking (OR 0.51, 95% CI 0.28 to 0.90, p = 0.021) reduced significantly | Significant reduction in alcohol use at population level in the control group (OR 0.71, 95% CI 0.57–0.88). Among patients with hypertension in the control group, alcohol use (OR 0.62, 95% CI 0.38 to 0.99, p = 0.044) reduced significantly | | No significant reduction in mean SBP and DBP at population level. No significant difference in SBP reduction among patients with hypertension. However, DBP decreased more in the control group than the intervention one (p = 0.028). Significant reduction in SBP and DBP in both intervention 2.75 mmHg (p = 0.001) and control groups 1.67 mmHg (p = 0.029) and larger reductions among those with hypertension in (intervention (SBP 4.82 mmHg, p < 0.001; DBP 7.55 mmHg, p < 0.001) and control (SDP 14.05 mmHg, p < 0.001; DBP 10.67 mmHg, p < 0.001) | | |

**TABLE 2 |** (*Continued*) Effect of community-based interventions on study outcomes, Community-based interventions for cardiovascular disease prevention in low- and middle-income countries: a systematic review, 2000–2019.

| Author | Knowledge | Physical activity | Diet | Smoking | Alcohol | Body weight | Blood pressure | Blood glucose | Lipid profile |
|---|---|---|---|---|---|---|---|---|---|
| Daivadanam, 2018 [33] | | | Significant, modest increase in fruit intake from baseline in the intervention arm (12.5%) and control (6.6%) but no difference between the groups. Significant increase in vegetable intake in intervention (13.99%) and control arms (13.66%) but no difference between the groups. Significant increase in vegetable procurement by 19% in the intervention arm compared to the control arm ($p = 0.008$). Monthly household consumption of salt, sugar and oil was greatly reduced in the intervention arm ($p < 0.001$) | | | | | | |
| Thankappan, 2018 [34] | | No statistically significant difference in leisure time physical activity between intervention and control groups (RR = 1.20, $p = 0.36$) | Intervention participants more likely to consume 5 or more servings of fruits and vegetables (RR 1.83, $p = 0.008$) compared with control | No statistically significant difference in tobacco use between intervention and control groups (RR = 0.79, $p = 0.11$) | Intervention participants had a greater reduction alcohol use (RR 0.77, $p = 0.018$) and the amount of alcohol consumed was lower among intervention participants ($p = 0.030$) | No statistically difference in waist circumference (mean difference: 0.67, $p = 0.14$) and waist to hip ratio (mean difference: 0.008, $p = 0.12$) between intervention and control groups | No statistically difference in systollic (mean difference: 1.22, $p = 0.13$) and diastollic blood pressure (mean difference: 1.12, $p = 0.06$) between intervention and control groups | Diabetes developed in 17.1% (79/463) of control participants and 14.9% (68/456) of intervention participants (relative risk [RR] 0.88, 95% CI 0.66 ± 1.16, $p = 0.36$). Compared with the control group, intervention participants had a greater reduction in IDRS score (mean difference: −1.50 points, $p = 0.022$) | No statistically significant difference observed among intervention and control groups for total cholesterol (mean difference: 0.01, $p = 0.79$), LDL cholesterol (mean difference: 0.02, $p = 0.73$) and triglycerides (mean difference: 0.96, $p = 0.07$) |
| Jayakrishnan, 2013 [35] | | | | Overall prevalence of smoking abstinence was 14.7% in the intervention and 6.8% in the control group (relative risk: 1.85, 95% CI: 1.05, 3.25). A total of 41.3% subjects in the intervention area and 13.6% in the control area had reduced smoking by 50% or more at the end of 12 months | | | | | |
| Ramachandran, 2006 [36] | | Physical activity showed an improvement from 41.7% to 58.8% in LSM and from 45.9% to 62.9% in LSM + MET group | Improvement in diet adherence registered in LSM group from 62.5% to 81.6% and LSM + MET groups from 62% to 81.9% | | | Significant increase in mean body weight in the control group at annual follow-up. Among intervention, increase was only in the lifestyle group at 24 months ($p = 0.035$) Mean waist circumference did not significant differ in any group relative to baseline values | | Significant relative risk reduction of 28.5%, $p = 0.018$ in the LSM group, 26.4% with MET group ($p = 0.029$) and 28.2% in LSM + MET group ($p = 0.022$) compared to control group | |
| Ramachandran, 2013 [37] | | Adherence to physical activity did not differ between the intervention and control groups | Intervention group had significantly greater diet adherence compared to control group ($p = 0.0442$) | | | No significant effect of intervention on BMI and waist circumference | No significant effect of intervention on systolic and diastollic blood pressure | 50 (18%) of men in intervention group developed type 2 diabetes compared with 73 (27%) in the control group (absolute risk reduction of 9%). Intervention thus reduced incidence of type 2 diabetes. ($p = 0.015$) | No significant effect of intervention on serum cholesterol and triglycerides. However, the effect on HDL cholesterol was significant |

**TABLE 2 |** (*Continued*) Effect of community-based interventions on study outcomes, Community-based interventions for cardiovascular disease prevention in low- and middle-income countries: a systematic review, 2000–2019.

| Author | Knowledge | Physical activity | Diet | Smoking | Alcohol | Body weight | Blood pressure | Blood glucose | Lipid profile |
|---|---|---|---|---|---|---|---|---|---|
| Nishtar, 2007 [38] | Significant positive changes in knowledge levels in intervention district compared with baseline levels in relation to a heart healthy diet (*p* < 0.001), beneficial level of physical activity (*p* < 0.001), causes of high blood pressure (*p* < 0.001) and heart attack (*p* < 0.001) and its causes (*p* < 0.001) and the effects of high blood pressure (*p* = 0.003) and active and passive smoking on health (*p* < 0.001). In control group, significant changes were for knowledge on: effects of smoking on health (*p* = 0.013). Significant differences were observed for all knowledge variables in favor of the ntervention | No changes were seen in the level of physical activity | Significant differences noted in consumption of two or more servings of vegetables per day within the intervention group (*p* < 0.001) and between the intervention and control group at the end of the intervention (0.020). No significant differences observed regarding consumption of five or more servings of fruits and vegetables, consumption of two or more fruit servings and type of oil/fat/ghee used for cooking | No significant changes observed comparing pre and post smoking pattern in both intervention and control groups. In comparing intervention and control sites post scores, there was a difference in usage of smokeless tobacco (*p* = 0.022) | | | | | |
| Jafar, 2009 [39] | | Median metabolic equivalent scores for physical activity increased in the HHE and GP group remained unchanged in the GP only and HHE only groups, and decreased in the no intervention group (*p* = 0.030 for difference among groups) | | Proportion of current smokers decreased from baseline to the last follow-up visit in all 4 groups (*p* < 0.001 in each). A 12.3% decrease in the HHE and GP group, an 11.9% decrease in the HHE-only and GP-only groups, and a 9.5% decrease in the no intervention group (*p* < 0.001 for difference among groups) | | Non-significant increase in mean BMI in all 4 groups but most marked in the no intervention group (0.20 [CI, −0.13 to 0.53] vs. 0.04 [CI, −0.38 to 0.05] in the HHE and GP group, 0.04 [CI, −0.19 to 0.28] in the GP-only group, and 0.09 [CI, −0.16 to 0.34] in the HHE-only group; *p* = 0.89 for difference among groups) | Decrease in SBP was significantly greater in the HHE and GP group (10.8 mm Hg [95% CI, 8.9–12.8 mm hg]) than in the GP-only (5.6 mm Hg (3.7–7.4), HHE-only (5.6 mm Hg (3.7–7.5) or no intervention groups (5.8 mm Hg [CI, 3.9–7.7 mm hg] in each; *p* < 0.001). DBP decreased in all groups but no significant difference in comparing groups (*p* = 0.27). Substantially greater proportion of patients achieved controlled blood pressure in the HHE and GP group than in other groups, *p* = 0.003 | | |
| Fottrell, 2019 [40] | Increases in ability to report one or more valid causes, symptoms, complications, and strategies for prevention and control of diabetes were observed in both intervention groups compared with control, with the effect consistently greatest in the PLA group (*p* < 0.01) | No significant difference in participants achieving an average of more than 150 min of physical activity per week among the intervention groups (PLA 0.83 (0.53–1.30; *p* = 0.418 and mHealth 0.98 (0.62–1.57; *p* = 0.945) when compared to control | No statistical difference in mean number of portions of fruits and vegetables consumed per day in both PLA (0.29 (0.10–0.69; *p* = 0.143) and m health (−0.19 (−0.53 to 0.15; *p* = 0.274) groups compared to control | | | No significant difference between the PLA and mHealth group compared control | | | Large reduction in combined prevalence of type 2 diabetes and intermediate hyperglycaemia in the PLA group compared with the control group, absolute reduction of 20·7% (95% CI 14·6–26·7) 2 years cumulative incidence of diabetes among intermediate hyperglycaemia cohort was significantly lower in the PLA group compared with control, absolute incidence reduction of 8·7% (3·5–14·0). No evidence of effect of mHealth on combined prevalence of intermediate hyperglycaemia and diabetes or the incidence of diabetes (1·02, 0·73–1·43) |

**TABLE 2** | (*Continued*) Effect of community-based interventions on study outcomes, Community-based interventions for cardiovascular disease prevention in low- and middle-income countries: a systematic review, 2000–2019.

| Author | Knowledge | Physical activity | Diet | Smoking | Alcohol | Body weight | Blood pressure | Blood glucose | Lipid profile |
|---|---|---|---|---|---|---|---|---|---|
| Nguyen, 2012 [41] | | In control group, physical inactivity only significantly changed in men and in intervention group, significant increase was in both men and women. When both groups were compared, significant decrease was observed in females | Prevalence of salty diets decreased significantly in intervention group and was unchanged in control group (*p* < 0.001) | Prevalence of smoking did not significantly change | Reduction in alcohol consumption in both intervention and control communities (*p* < 0.01) | Waist circumference and waist hip ratio significantly increased in both groups (*p* < 0.01). Weight and BMI was unchanged in intervention group but significantly reduced in control community for BMI in both sexes (*p* < 0.05) and weight in women (*p* < 0.01). Intervention group registered significant increase in weight, BMI and waist circumference in women and in men, BMI also increased | SBP reduced in both groups while DBP reduced only in the intervention group (*p* < 0.01). Comparing changes over time between groups, SBP and DBP significantly decreased in intervention group. SBP vs DBP (3.3 and 4.7 mmHg in women vs. 3.0 and 4.6 mmHg in men, respectively) | | |
| Chandraratne, 2019 [42] | | No significant difference in odds of engaging in recommended levels of leisure time physical activity between groups at follow-up (*p* = 0.15) and mean sedentary minutes (*p* = 0.94) | Significantly higher odds of consuming one or more serving/day of fruits (OR 1.71, 1.10–2.65, *p* = 0.02) and significantly lower odds of consuming two or more snacks per day (OR 0.32, 0.21–0.48, *p* < 0.001) in intervention than control group. No significant difference in the consumption of vegetables or sugar sweetened beverages between groups | No significant difference in proportion of smokers in both groups | No significant difference in proportion that consumed more than two drinks per day for men and one drink for women | Significantly lower mean body weight (61.8 kg [12.7]) and BMI (24.4 kg/m2) in intervention than control group (body weight 64.0 kg [12.8], BMI 25.5 kg/m2 (4.7)). Mean difference: Body weight –2.83 kg (–3.31 to –2.35) and BMI (–1.12 kg/m2 (–1.32 to –0.94)). More pronounced difference among overweight persons (body weight –3.69 kg and BMI –1.50 kg/ m2, *p* < 0.001) | No significant change in blood pressure from baseline to endline in both groups including among hypertensives. Mean difference in SBP (–0.88) and DBP (–0.94) between intervention and control groups was also not significant | | |
| Joshi, 2019 [43] | | | | Proportion of individuals who use smokeless tobacco significantly declined between baseline and 12 months in both intervention and control groups and difference between groups was significant (*p* = 0.001). Proportion of individuals who smoked declined by 4.1% in intervention arm and by 2.6% in control arm but the difference between the two groups was not statistically significant | | | Significant decline in SBP (mmHg) from baseline in both groups-controls 130.3 ± 21 to 128.3 ± 15; intervention 130.3 ± 21 to 127.6 ± 15 (*p* < 0.01 for before and after comparison) but there was no difference between the two groups at 12 months (*p* = 0.18) | | |
| Gunawardena, 2016 [44] | | The intervention group had a significantly higher odds of engaging in adequate physical activity than the control group (AOR 3.25 (95% CI 1.87–5.62). Leisure-time activity showed a significant difference between the groups (*p* < 0.0001). A significantly greater increase in the number of daily steps in intervention than the control group (*p* < 0.0001) | No significant difference in individual-level food consumption between the two groups including fruits and vegetables consumption after the intervention, but the intervention group showed a significant decrease in household-level purchase of biscuits and ice cream than the control group (*p* < 0.0001 and *p* = 0.03, respectively) | | | Intervention group had a significantly lower mean of weight and BMI than did control group (*p* < 0.0001); mean (95% CI) effect between groups was –2.49 (–3.38 to –1.60) kg and –0.99 (–1.40 to –0.58) kg/m2 | | | |

**TABLE 2 |** (*Continued*) Effect of community-based interventions on study outcomes, Community-based interventions for cardiovascular disease prevention in low- and middle-income countries: a systematic review, 2000–2019.

| Author | Knowledge | Physical activity | Diet | Smoking | Alcohol | Body weight | Blood pressure | Blood glucose | Lipid profile |
|---|---|---|---|---|---|---|---|---|---|
| Joshi, 2012 [45] | No statistically significant effect on knowledge about 6 lifestyle factors affecting CVD risk ($p = 0.15$) | No significant difference in mean number of days of light and medium/heavy physical activity ($p = 0.78$) | Individuals in the villages who received a health promotion intervention were significantly more likely to avoid consumption of oily foods ($p = 0.01$). No change in mean no. of days eat fruit, green leafy vegetables and salt consumption | No significant change in proportion not currently using tobacco ($p = 0.74$) | | No change in mean body mass index, mean waist circumference | No significant change in mean systolic, diastolic between intervention and control groups following intervention | | |
| Lu, 2015 [46] | Improvements in hypertension-related knowledge score registered in all groups but greatest in group 3, then 2 and 1 | Regular physical activity increased in all groups progressively from group 1 (self-learning reading) to 2 (regular didactic lecture) and highest in 3 (interactive education workshops) | Adherence to appropriate salt intake was progressively greater from group 1 to 3 | No significant difference in proportion of current smokers in all groups | No significant difference in proportion of alcohol drinkers in all groups | BMI decreased significantly in group 1 (self-reading) and 3 (interactive education workshop) - largest reduction and waist and hip ratio decreased significantly only in group 3. The difference across groups was statistically significant | Proportion of subjects with normalized BP increased in group 2 (41.2%–63.2%, $p < 0.001$) and more substantially in group 3 (40.2%–86.3%, $p < 0.001$). No significant change in group 1 | | No significant differences in serum total cholesterol and LDL concentration among groups. Serum total cholesterol decreased significantly in group 2 and 3 and not 1. Serum triglycerides concentration decreased in group 1 only. HDL concentrations increased significantly in group 1 and 3 but not 2. Fasting LDL concentrations increased significantly in group 1, didn't change significantly in group 2 and decreased significantly in group 3 |
| Lv, 2014 [47] | Tobacco related knowledge significantly improved in intervention compared to control. Diet and physical activity knowledge and beliefs decreased in intervention compared to control | The metabolic equivalent of physical activity significantly increased in intervention ($p = 0.023$) compared to control ($p = 0.201$) | The fruit and vegetable consumption score significantly increased in both the intervention (24.84 vs 25.97, $p = 0.036$) and comparison (24.25 vs 26.67, $p < 0.001$) areas but difference between the two not significant | A statistically significant decline in the prevalence of smoking observed in intervention (25.2% vs 18.7%, $p < 0.001$) compared with the comparison area (18.0% vs 16.4%, $p = 0.343$). Statistically significant difference in prevalence also observed among men | | | | | |
| Chao, 2012 [48] | | Intervention group demonstrated significant improvement in physical activity duration per week | Intervention group demonstrated significant improvement in diet score compared to control group | | | Waist to hip ratio significantly reduced in the intervention group compared to controls | Systollic blood pressure significantly reduced in the intervention compared to controls. No difference observed in diastollic blood pressure between groups | Fasting blood sugar significantly reduced in the intervention group compared to controls | No difference observed in lipidemia patients triglyceride between groups |
| Huang, 2011 [49] | Statistically significant changes in hypertension knowledge ($p < 0.05$) | The percentage of participants physically active increased from 43.3 to 59.7% in the intervention group and 43.7–70.2% in the control group, both statistically significant | Participants in intervention group exhibited a significantly greater improvement in dietary habits including reducing salty food intake (13.6% vs. 21.7%), fat intake (22.9% vs. 31.9%) comparison with those in the control group. Consumption of pickled food reduced in intervention and increased in control group though not statistically significant. Reduction in consumption of pickled food registered in both groups but difference not significant ($p = 0.641$) | Smoking reduced in the intervention group (29.5–26.0%) and increased in the control group (23.9–27.0%) and the difference was not statistically significant | Intervention group participants exhibited a significantly greater improvement in alcohol consumption compared with control group (9.6%) vs. 18.0%, $p < 0.05$) | | A significant reduction in the prevalence rate of hypertension in only the intervention group, which was from 35.4 to 22.5%. No change in the prevalence of hypertension in the control group. Significant increase in treatment ($p < 0.05$) and control rates of hypertensive patients in both groups ($p < 0.05$) and between group in favor of the control group ($p < 0.05$) | | |

**TABLE 2 |** (*Continued*) Effect of community-based interventions on study outcomes, Community-based interventions for cardiovascular disease prevention in low- and middle-income countries: a systematic review, 2000–2019.

| Author | Knowledge | Physical activity | Diet | Smoking | Alcohol | Body weight | Blood pressure | Blood glucose | Lipid profile |
|---|---|---|---|---|---|---|---|---|---|
| Zhang, 2018 [50] | | Comparing intervention and control groups, adherence to physical activity was significant at both 12 and 24 months | Comparing intervention and control groups, adherence to high diet score was significant at both 12 and 24 months | Adherence to non-smoking among intervention compared to control was not significant at both 12 and 24 months | Adherence to moderate alcohol use among intervention compared to control was not significant at both 12 and 24 months | Average waist circumference lower in intervention group but not statistically significant both at 12 and 24 months | Average systolic and diastollic BP significantly lower in intervention group compared to control both at 12 and 24 months | Fasting plasma glucose significantly lower in intervention group compared to control both at 12 and 24 months | Total cholesterol significantly lower in intervention group than control both at 12 and 24 months |
| Ibrahim, 2016 [51] | | Greater proportion of participants from the Co-HELP group met the clinical recommended target physical activity of >600 METS/min/wk (60.7% vs 32.2%, $p < 0.001$) compared to the usual care group | Intervention group showed a greater percentage of participants (13.9%) who met the dietary aims (to reduce 20 ± 25 kcal/kg energy intake) as compared to usual care group (9.6%) but not statistically significant | | | Greater proportion of intervention participants met the clinical recommended target of 5% or more weight loss from the initial weight (24.6% vs. 3.4%, $p < 0.001$). Change in waist circumference was −2.44 cm (−4.75 to −0.12, $p < 0.05$) | At 12 months, SBP reduced in intervention compared to control group but not significant (−1.71 (−3.97 to 0.56)). DBP changed by −2.63 mmHg (−3.79 to −1.48, $p < 0.01$) compared to the control group | Analysis of between-groups at 12 months (mean difference, 95% CI) revealed that the Co-HELP participants' mean fasting plasma glucose reduced by −0.40 mmol/L (−0.51 to −0.28, $p < 0.001$), 2 h post glucose by −0.58 mmol/L (−0.91 to −0.24, $p < 0.001$), HbA1C by −0.24% (−0.34 to −0.15, $p < 0.001$) | HDL cholesterol increased by 0.12 mmol/L (0.05–0.19, $p < 0.01$), compared to the usual care group |
| Mcalister, 2000 [52] | | | | Cessation rates were 7–26% in pitkäranta and 1–2% in the comparison area ($p < 0.05$) | | | | | |
| Aung, 2019 [53] | | | | Intervention participants (25.62%) achieved a significantly higher smoking cessation rate than the control participants (11.32%), adjusted analysis (AOR 2.95, $p < 0.001$) | | | | | |
| Latina, 2020 [54] | | No significant difference between the two groups in exercise | | No significant change in tobacco use between groups | | | | At one-year of follow-up, the overall FBS was significantly different between the peer group intervention and control groups [9.1 (SD 2.7) vs. 8.5 (SD 2.6), $p = 0.028$ | |
| Anthony, 2015 [55] | | No significant difference between intervention and control in change in physical activity | The proportion eating five portions of fruit and vegetables increased in intervention compared to control group (6.9% vs. 1.5%, $p < 0.001$). Salt intake increased in both groups more in the control compared to the intervention group ($p = 0.014$) | Prevalence of tobacco use significantly reduced in men (6.0% vs. 2.6%, $p < 0.001$) in intervention compared with control. In women, tobacco use slightly increased in both groups with no difference | | No significant difference between intervention and control in change in overweight | | | |
| Neupane, 2018 [56] | | No significant differences between intervention and control groups at follow-up in proportions of people who had low physical activity (OR = 0.77, 95% CI 0 24–2.45) | No significant differences between the intervention and control groups at follow-up in proportions of people who consumed 5 g or more of salt each day (0.80, 0.56–1.14) and ate less than five servings of fruit and vegetables each day (OR = 1.09, 95% CI 0 38–3.13) | No significant differences between the intervention and control groups at follow-up in proportions of people who smoked daily (OR 0.79, 95% CI 0 46–1.37) | No significant differences between groups at follow-up in proportions of people who consumed harmful amounts of alcohol (OR 1.07, 95% CI 0 61–1.90) | | Mean SBP at 1 year was significantly lower in the intervention group than in the control group for all cohorts: (Difference −2.28 mm Hg (95% CI −3.77 to −0.79, $p = 0.003$) for normotensive participants, −3.08 mm Hg (−5.58 to −0.59, $p = 0.015$) for prehypertensive, and −4.90 mm Hg (−7.78 to −2.00, $p = 0.001$) for hypertensive participants | | |

could have provided recent studies. However, such abstracts sometimes do not contain adequate information and their results may be inconclusive. The heterogeneity of the outcome measures also meant that a meta-analysis was not possible. Nevertheless, this systematic review provides evidence on the effectiveness of community-based interventions for CVD prevention in LMICs and their effectiveness in improving knowledge and uptake of healthy lifestyles in addition to changing metabolic risk factors essential for CVD prevention.

## Conclusion

This review found several community-based interventions implemented for CVD prevention in LMICs which significantly influenced knowledge about CVD and risk factors, and changes in physical activity and dietary behaviors for CVD prevention. However, evidence on reducing smoking and alcohol consumption were inconsistent necessitating further research. Regarding the CVD metabolic risk factors, community-based interventions significantly impacted blood pressure and blood sugar measurements. The most effective interventions utilized community mobilization, health education and information sharing, individual or group counseling, and trainings of providers. Evidence from this review can inform policy makers in decision-making and prioritizing evidence-based interventions for CVD prevention in LMICs.

## AUTHOR CONTRIBUTIONS

RN, RW, HB and GM conceptualized this review. RN, HH, RW, DM, FN, SA, HB, GM contributed to the review protocol and were

involved in conducting the review and supervising the process. RN prepared the original draft of the manuscript. HH, RW, DM, FN, SA, HB, GM thoroughly reviewed and edited the manuscript. All authors read and approved the final version of the manuscript.

## FUNDING

This work was supported by the SPICES project in Uganda which received funding from the European Commission through the Horizon 2020 research and innovation action grant agreement No 733356 to implement and evaluate a comprehensive CVD prevention program in five settings: a rural and semi-urban community in a low-income country (Uganda), middle income (South Africa) and vulnerable groups in three high-income countries (Belgium, France and United Kingdom). The funder had no role in the design, decision to publish, or preparation of the manuscript.

## CONFLICT OF INTEREST

The authors declare that the research was conducted in the absence of any commercial or financial relationships that could be construed as a potential conflict of interest.

## ACKNOWLEDGMENTS

The authors wish to thank Isaac Ddumba for his contribution to the review protocol.

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

***PHR is edited by the Swiss School of Public Health (SSPH+) in a partnership with the Association of Schools of Public Health of the European Region (ASPHER)+***

