## [Reviewer comments · Public Health Reviews]

Peer Review Report

Review Report on Community-based interventions for cardiovascular disease prevention in low- and middle-income countries: a systematic review

Systematic Review, Public Health Rev

Reviewer: Marta Fadda

Submitted on: 16 Mar 2021

Article DOI: 10.3389/phrs.2021.1604018

EVALUATION

Q 1 Please summarize the main theme of the review.

The article reports on a review whose aim is to summarize evidence on the effectiveness of community-based interventions for cardiovascular disease prevention in low- and middle- income countries.

Q 2 Please highlight the limitations and strengths.

Strengths

The article has important findings related to an understudied geographical area (LMICs) and provides up-to-date and comprehensive evidence on the effectiveness of community-based interventions on cardiovascular disease prevention in these settings.

Limitations

The discussion could be improved by addressing specific features on the studies included in the review

Q 3 Please provide your detailed review report to the authors, structured in major and minor comments.

- Please address the lack of studies from low-income countries in the discussion and the higher prevalence of studies from Asian countries
- When discussing interventions that were effective in changing either the primary or the secondary outcomes, please address whether effectiveness changed based on type of interventions (training, community mobilization, health education and consultation delivered through campaigns, group meetings, workshops, use of mobile technologies and of health workers, community health workers or peers), on having more than one intervention, and whether this is line with evidence from high income countries.

PLEASE COMMENT

Q 4 Is the title appropriate, concise, attractive?

YES

Q 5 Are the keywords appropriate?

YES

Q 6 Is the English language of sufficient quality?

YES

Q 7 Is the quality of the figures and tables satisfactory?

Yes.

Q 8 Does the reference list cover the relevant literature adequately and in an unbiased manner?

YES

Q 9 Does this manuscript refer only to published data? (unpublished data is not allowed for Reviews)

Yes.

Q 10 Does the manuscript cover the issue in an objective and analytical manner

Yes.

Q 11 Was a review on the issue published in the past 12 months?

No.

Q 12 Does the review have international or global implications?

YES

QUALITY ASSESSMENT

Q 13 Quality of generalization and summary

Q 14 Significance to the field

Q 15 Interest to a general audience

Q 16 Quality of the writing

REVISION LEVEL

Q 17 Please take a decision based on your comments:

Minor revisions.

---

## [Reviewer comments · Public Health Reviews]

Peer Review Report

Review Report on Community-based interventions for cardiovascular disease prevention in low- and middle-income countries: a systematic review

Systematic Review, Public Health Rev

Reviewer: Pablo Perel

Submitted on: 09 Apr 2021

Article DOI: 10.3389/phrs.2021.1604018

EVALUATION

Q 1 Please summarize the main theme of the review.

This review summarized 27 studies evaluating the effectiveness of community-based interventions for primordial or primary prevention of cardiovascular disease in low- and middle- income countries.

Q 2 Please highlight the limitations and strengths.

The main limitation is that there is a lack of clear link between the risk of bias evaluation and the presentation of the studies results therefore it is difficult to assess the strength of the evidence for the results presented. There is an too much emphasis in the reporting of a long list of "statistically significant" or not significant results. The review would be more useful if a clearer link is made between the strength of the evidence and the findings. Also, with the type of complex interventions that is the focus of the review it would be informative to include issues beyond effectiveness (e.g. aspects which can facilitate or hinder implementation)

Q 3 Please provide your detailed review report to the authors, structured in major and minor comments.

Major comments

It would be important to make a specific link between the characteristics of the studies in terms of risk of bias and the finding of the studies. The two sections are not linked now, and the risk of bias is quite short. For a better understanding of the finding, I would suggest putting less emphasis on "statistically" or " not statistically" results and refer to the strength of evidence which should consider on top of the potential role of chance (statistical tests) the potential role of bias due to study design . It would also be useful to include aspects that were reported in process evaluation of the studies that can hinder or facilitate implementation as this information might be quite useful when considering the implementation of the interventions to different settings.

Also, although the authors include as one of the key research questions "What community-based interventions and strategies have been implemented for CVD prevention in LMICs?" I do not think that actually the review answers this question as it only focussed on research studies (not all community-based interventions and strategies) and because even within the research studies the eligibility criteria and the language limitation means that this question is not fully answered by the review.

Minor comments

As the main review question is related to "community based interventions" it would be useful to include a definition of this term in the introduction.

If possible structure the eligibility criteria as PICO and Include a rationale for the selected criteria (why 150 participants , 9 months, 40% attrition, etc)

Because of the nature of the review and the lack of meta-analysis it is not clear why outcomes should be predefined as "primary" or "secondary", I would suggest to display all the outcomes measured (perhaps include a heat map?) rather than artificially dividing in primary and secondary ones.

Would be useful to include the search terms in the search strategy (not sure if I missed them but could not find them)

I would suggest to expand the discussion and details of risk of bias (and, as mentioned before, link specifically the bias with the result section)

It would also be interesting to report how many of the studies clearly described the intervention (a major limitation in complex intervention studies)

For presenting the overall results in the “Variety of community-based interventions” section it might make the reading easier if the authors structure this on a table or figure What (content of the intervention) , where (setting, including clinical and geographical setting) and how (who were involved in delivering the intervention) or any other format that authors might consider appropriate.

As mentioned before I would suggest to put less emphasis on "positive" or significant studies in the conclusion , also highlights research needs in relation to current methodological limitations

PLEASE COMMENT

Q 4 Is the title appropriate, concise, attractive?

Yes

Q 5 Are the keywords appropriate?

I would add prevention

Q 6 Is the English language of sufficient quality?

Yes

Q 7 Is the quality of the figures and tables satisfactory?

No.

Q 8 Does the reference list cover the relevant literature adequately and in an unbiased manner?

I suggest to include latest figure of GBD study (2019 instead of 2016)

Reference 5 is not so recent suggest to edit accordingly.

Q 9 Does this manuscript refer only to published data? (unpublished data is not allowed for Reviews)

Yes.

Q 10 Does the manuscript cover the issue in an objective and analytical manner

Yes.

Q 11 Was a review on the issue published in the past 12 months?

No.

Q 12 Does the review have international or global implications?

The topic is highly relevant but the way is written now with a long list of "statistically" or "not statistically" "significant" results is not very informative

QUALITY ASSESSMENT

Q 13 Quality of generalization and summary

Q 14 Significance to the field

Q 15 Interest to a general audience

Q 16 Quality of the writing

REVISION LEVEL

Q 17 Please take a decision based on your comments:

Major revisions.

---

## [Reviewer comments · Public Health Reviews]

Peer Review Report

Review Report on Community-based interventions for cardiovascular disease prevention in low- and middle-income countries: a systematic review

Systematic Review, Public Health Rev

Reviewer: Louise Hartley

Submitted on: 19 Mar 2021

Article DOI: 10.3389/phrs.2021.1604018

EVALUATION

Q 1 Please summarize the main theme of the review.

The review aimed to provide a comprehensive evidence base as to the effectiveness of community-based interventions for CVD prevention so as to support the design of strategies in LMICs for the prevention of CVD

Q 2 Please highlight the limitations and strengths.

Strengths

- The review was conducted in accordance with recognised standards and the protocol registered in PROSPERO
- Clear research questions
- Used validated quality assessment checklists
- Provides a PRISMA flowchart for the flow of studies through the review
- Important research topic

Limitations

- Not clear that the studies included in this review are a subset of a larger review
- No form of analysis makes the review difficult to follow and no reason provided for why no analysis has been performed in the methods/results sections.
- The results section as written is hard to follow. It is not clear which types of community interventions are included for each outcome and to what each of the interventions is being compared.
- The study characteristics table states country of the study but not if low or middle income
- Did not include conference abstracts so might have missed recent studies that have yet to be published as a full manuscript
- Did not search the Cochrane Database of Systematic Reviews which might have included relevant SLRs to hand search
- No justification for timeframe of review (2000-2019) nor why limited to English language only
- Unsure of the term snowball sources
- The literature search results section is not clear. Without the PRISMA flowchart it would not be clear as to how many studies were included/excluded at each level of screening.
- First section of study setting is unclear. States 32 articles were extracted but 27 studies included in the review. Need to clarify that 5 studies were secondary publications
- Errors made with regards to HDL cholesterol reporting. If the intervention is effective then HDL should increase but in the Lipids section the authors include lowering HDL with lowering LDL, total cholesterol, and triglycerides. It is unclear from this section which interventions increased HDL and lowered other lipid levels
- The discussion section does not clearly provide a summary of the main findings and few limitations of the review are reported

Q 3 Please provide your detailed review report to the authors, structured in major and minor comments.

Major Comments

- Reasons need to be given as to why no analysis has been conducted.
- The results section is quite text heavy, and no measures of effect are reported in the results. Only frequency counts of the number of studies reporting the intervention as better than the control and so on. For each outcome, a forest plot could be created to show the treatment effect in each study for the intervention vs. the control with no pooled

estimates. This would provide a good visual to see which community interventions are better than controls and which are not better. Each forest plot could also have sub-groups for low-income countries and middle-income countries so it can be seen if the effectiveness of community interventions differs according to this.

- The current results section is also difficult to follow. It may be clearer if for each outcome results are split into those from RCTS and those from other study designs. For example, body weight could have the results from RCTs followed by the results from before and after studies. It may also be useful to split the outcomes by low- and middle-income status.
- Please ensure that results for HDL are reported correctly. An increase in HDL is good.

Minor Comments

- Provide a reason in the methods as to why English language only articles were included and why the timeframe from 2000-2019 was chosen
- Study setting section on page 6 line 164: clarification needed to say that 5 articles were secondary references, and 27 articles were primary references. So, 27 studies were included in total from 32 publications
- Please clarify “snowball sources” on line 115
- No conference abstracts were included but these may have included recent relevant studies. Please list this as a limitation in the discussion
- Please add whether the study country was classed as low or middle income to the study characteristics table
- Please add to the text how many studies were classified as low or middle income
- Needs to be made clearer that the included studies in this research are a subset of studies from a larger systematic review. This should be included in the methods section.
- Please add a figure for risk of bias to Risk of Bias section (page 6 line 182)
- Please clarify whether the PICOS criteria was used for the eligibility criteria in the review. This paragraph could also be made clearer in the manuscript by having separate paragraphs for each of the PICOS criteria (Page 4 line 80)
- The sentence on line 135-136 starting “To avoid double counting,”. is currently not clear. Could this please be re-written.
- Please add comparators to table 1 and a description of comparators in the study characteristics text on page 7
- The year of publication section from line 197 onwards can be deleted as it does not provide useful information for the systematic review
- Line 285-286 states “...smoking prevalence in favour of the intervention compared to the intervention group”. Please adjust this sentence to say control/comparison group
- The sentence starting on line 319 states “...found significant reductions inand prevalence of diabetes and intermediate hyperglycaemia [41]” while the sentence starting on line 323 states “...the m-health intervention arm did not influence the combined prevalence of intermediate hyperglycaemia and diabetes...[41]”. This is confusing since it is the same reference with conflicting results. Please provide clarification on this study’s findings
- The literature search results section starting on page 5 line 154 is unclear. This needs to be re-written so that without the PRISMA diagram the number of included and excluded studies at level 1 and level 2 screen can be easily identified.
- Provide a summary paragraph in the discussion of the main review findings and further report on the limitations of this systematic literature review

PLEASE COMMENT

Q 4 Is the title appropriate, concise, attractive?

Yes, the title is appropriate and describes the study well

Q 5 Are the keywords appropriate?

The key words are appropriate but I would maybe add the word "prevention"

Q 6 Is the English language of sufficient quality?

Unfortunately, some of the paragraphs and sentences are difficult to understand and will need clarification

Q 7 Is the quality of the figures and tables satisfactory?

Yes.

Q 8 Does the reference list cover the relevant literature adequately and in an unbiased manner?

Yes

Q 9 Does this manuscript refer only to published data? (unpublished data is not allowed for Reviews)

Yes.

Q 10 Does the manuscript cover the issue in an objective and analytical manner

Yes.

Q 11 Was a review on the issue published in the past 12 months?

No.

Q 12 Does the review have international or global implications?

No answer given.

QUALITY ASSESSMENT

Q 13 Quality of generalization and summary

Q 14 Significance to the field

Q 15 Interest to a general audience

Q 16 Quality of the writing

REVISION LEVEL

Q 17 Please take a decision based on your comments:

Major revisions.